# Robust Adversarial Attacks Against Unknown Disturbances via Inverse Gradient Sample

**Zhaoyang Zhang**[1], **Shen Wang**[1], **Runze Liu**[1], **Guopu Zhu**[1], **Fanghui Sun**[1],
**Ye Lu**[1], **Zeyue Wang**[2], **Yihan Yan**[1,⋆]
[1] Faculty of Computing, Harbin Institute of Technology, Harbin 150001, China
[2] Bio-vision System Laboratory, Shanghai Institute of Microsystem and Information
Technology, Chinese Academy of Sciences, Shanghai 200050, China.
`yan.office366.m@hit.edu.cn`

## Abstract

Adversarial attacks have achieved widespread success in various domains, yet existing methods suffer from significant performance degradation when adversarial examples are subjected to even minor disturbances. In this paper, we propose a novel and robust attack called IGSA (**I**nverse **G**radient **S**ample-based **A**ttack), capable of generating adversarial examples that remain effective under diverse unknown disturbances. IGSA employs an iterative two-step framework: (i) inverse gradient sampling, which searches for the most disruptive direction within the neighborhood of adversarial examples, and (ii) disturbance-guided refinement, which updates adversarial examples via gradient descent along the identified disruptive disturbance. Theoretical analysis reveals that IGSA enhances robustness by increasing the likelihood of adversarial examples within the data distribution. Extensive experiments in both white-box and black-box attack scenarios demonstrate that IGSA significantly outperforms state-of-the-art attacks in terms of robustness against various unknown disturbances. Moreover, IGSA exhibits superior performance when attacking adversarially trained defense models. Code is available at `https://github.com/nimingck/IGSA`.

## 1 Introduction

Extensive research demonstrates that deep neural networks (DNNs) are highly vulnerable to adversarial examples Szegedy (2013); Papernot et al. (2017); Kurakin et al. (2018). The emergence of more threatening adversarial examples has the potential to stimulate advances in secure machine learning Liu et al. (2016); Leino et al. (2021); Zhu et al. (2023b). To be genuinely threatening in practice, an adversarial example should satisfy three key properties: (i) transferability, ensuring its effectiveness in black-box scenarios; (ii) stealthiness, enabling it to evade standard detection mechanisms; and (iii) robustness, allowing it to retain attack effectiveness under various disturbances.

A widely studied category of adversarial attacks is the white-box attack Goodfellow et al. (2014); Carlini & Wagner (2017); Kurakin et al. (2018), which assumes full access to the target model's parameters and architecture. While effective in theory, this assumption rarely holds in practice, limiting their real-world relevance. A more practical alternative is the transfer-based black-box attack Papernot et al. (2016); Wu et al. (2020), where adversarial examples generated on surrogate models are applied to unknown target models. Yet, recent evidence Liu et al. (2024); Li et al. (2022); Xie et al. (2017) suggests that existing transfer attacks are highly brittle: even minor disturbances can result in the effectiveness of the attack, especially in targeted attacks, as shown in Fig 1. The fragility of adversarial examples naturally limits their attack success rate in applications.

In this paper, we propose a novel adversarial attack framework designed to enhance the robustness of adversarial examples against various (including unseen) disturbances. It adopts an iterative two-step procedure. First, disturbances are sampled from a prior distribution and mapped into a specified disturbance distribution, which relatively represent diverse and realistic disturbances. Second, the adversarial example is optimized to maintain its effectiveness under the sampled disturbance.

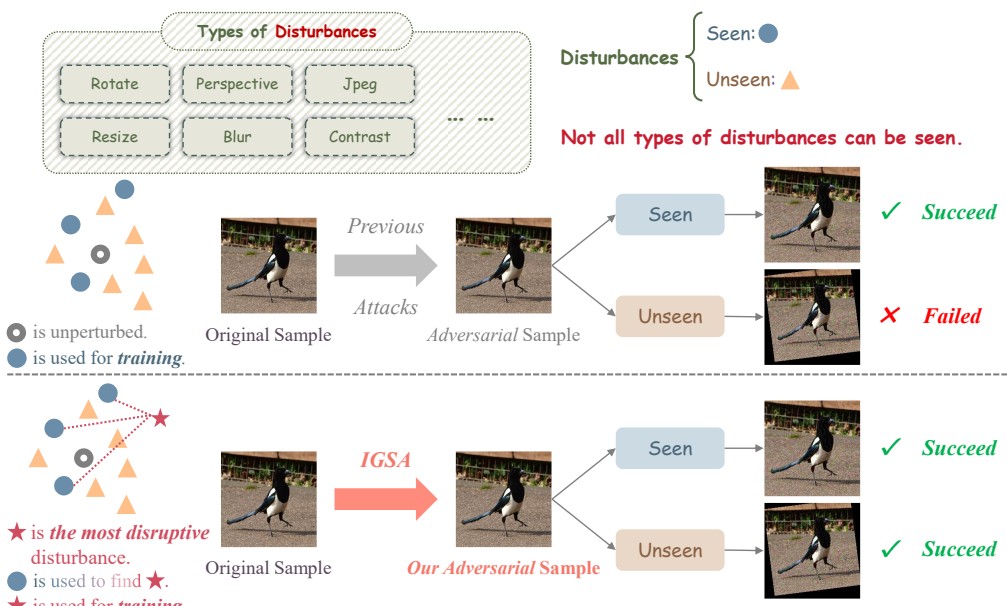

Figure 1: Robustness of adversarial attacks under various disturbances. Existing adversarial examples degrade under unseen disturbances. Our proposed IGSA enhances robustness against both seen and unseen disturbances.

The design of an appropriate mapping function in our robust attack framework raises three key challenges. (i) **Sampling Coverage Limitation:** When the disturbances are insufficiently sampled, adversarial examples may still fail under unseen disturbances. (ii) **Distribution Mismatch:** If the distribution of disturbances used for training differs from the actual distribution of real-world disturbance, adversarial examples may lose their effectiveness in practice. (iii) **Transferability Consideration:** In black-box scenarios, adversarial examples must remain transferable across models, necessitating explicit integration of transferability into training.

To address these challenges, we propose the **I**nverse **G**radient **S**ample-based **A**ttack (IGSA). In contrast to prior approaches that rely on random disturbance sampling during training Athalye et al. (2018), IGSA employs inverse gradient sampling to identify the most disruptive disturbances. This mechanism effectively mitigates the failure of adversarial examples under unseen or real-world noise. Theoretical analysis further shows that IGSA achieves over $10^8$ times higher efficiency in approximating such disturbances compared to random sampling. Moreover, IGSA actively explores flat regions of the loss landscape, a strategy recently shown Ge et al. (2023) to substantially enhance transferability of adversarial examples.

By analyzing the impact of data likelihood on robustness of adversarial examples, we evaluate IGSA under distribution mismatch. Extensive experiments show that models exhibit high confidence and robustness on clean samples from the natural distribution Liu et al. (2025). Theoretical analysis reveals that IGSA preserves a high likelihood of adversarial examples under the natural data distribution. This enables IGSA to generate adversarial examples that are both robust and resistant to defenses. Our main contributions are summarized as follows:

- We propose a robust attack framework that iteratively samples disturbances from a prior distribution and refines adversarial examples under these disturbances. The framework can be applied to any existing attack, enabling effective resilience against diverse disturbances.

- We introduce IGSA to address three key challenges in the robust attack framework. Theoretical analysis shows that IGSA improves the data likelihood of adversarial examples, enhancing its robustness against disruptions and defenses.

- Extensive experiments demonstrate that IGSA maintains high data likelihood during training, generating visually natural adversarial examples with strong attack success. Furthermore, the results show that IGSA outperforms state-of-the-art methods in both robustness and transferability.

## 2 RELATED WORK

### 2.1 BLACK-BOX ADVERSARIAL ATTACK

Black-box adversarial attacks are typically categorized into query-based Cheng et al. (2019); Dong et al. (2021); Shi et al. (2022) and transfer-based approaches Xie et al. (2019); Wang & He (2021); Wang et al. (2021); Jin et al. (2023); Chen et al. (2023); Wang et al. (2024b;a). Query-based methods estimate gradients by iteratively querying the target model, but they often require excessive queries, limiting their practicality under query constraints. In contrast, transfer-based methods generate adversarial examples on surrogate models and transfer them to the target model. To enhance transferability, prior work has explored diverse strategies, including momentum integration Wang et al. (2024a), input transformations Xie et al. (2019); Wang et al. (2021), model-specific strategies Jin et al. (2023); Wang et al. (2024b), and gradient ensembling Chen et al. (2023).

Despite their effectiveness, many of these attacks fail under even basic input transformations Xie et al. (2017); Xu (2017); Li et al. (2022); Liu et al. (2024), revealing a lack of robustness in real-world scenarios. To mitigate this issue, researchers have proposed several strategies. The Expectation over Transformation (EOT) framework Athalye et al. (2018) incorporates data augmentation during training to simulate distributional disturbances. Other techniques, such as gradient smoothing Wang et al. (2023), physical-world disturbances Eykholt et al. (2018), affine-invariant estimation Xu et al. (2020), and margin maximization Luo et al. (2018), further enhance attack stability. Nevertheless, these methods are largely heuristic, exhibit limited generalization to diverse disturbances, and lack theoretical performance guarantees.

### 2.2 DEFENSE METHODS

The number of existing defense methods far exceeds that of adversarial attacks, as stronger attacks continually motivate the development of more effective defenses. Broadly, defenses can be categorized into adversarial training-based and input transformation-based approaches. Adversarial training defenses Tramèr et al. (2017); Liu et al. (2020a); Jiang et al. (2023) enhance robustness by incorporating adversarial examples during optimization, but are computationally intensive. Input transformation defenses, on the other hand, attempt to neutralize perturbations before feeding them into the model through techniques such as JPEG compression Dziugaite et al. (2016), image scaling Xu (2017); Zheng et al. (2023), or randomized transformations Xie et al. (2017). Some methods further employ denoising networks to purify inputs while preserving accuracy Hong & Lee (2024); Ning et al. (2024), though their effectiveness is often restricted to specific attack types. These methods are attractive in practice, as they do not require modifications to the model architecture or additional training cost, making them both efficient and easy to deploy.

## 3 METHODOLOGY

### 3.1 PRELIMINARY: ROBUST ADVERSARIAL ATTACK FRAMEWORK

Given an original sample $x \in \mathbb{R}^m$ and a target model $f : \mathbb{R}^m \to \mathbb{R}^k$, the goal of adversarial attacks is to find a minimal perturbation $\delta$ such that the perturbed sample $x + \delta$ is misclassified by the model into a specified target class $t$, i.e., $f(x + \delta) = t$.

In black-box settings, optimizing $\delta$ is particularly challenging because adversarial examples may be subjected to additional disturbances before being processed by the target model. These disturbances can arise from various sources, such as secondary data acquisition, client-side preprocessing, or built-in defense mechanisms. To enhance the robustness of adversarial examples against disturbance, we propose a novel robust attack framework. The framework operates in two stages, aiming to generate perturbations that remain effective under diverse and potentially unseen disturbances:

**Step 1: Sampling disturbance**
*We first sample a set of initial disturbances $\phi$ from a prior distribution $\mathcal{B}$. These disturbances are then translated to $h(\phi, x + \delta)$ by a mapping function $h$, given the current adversarial example $x + \delta$.*
**Step 2: Optimizing adversarial examples**
*We apply the disturbed sample $x + \delta + h(\phi, x + \delta)$ to a surrogate model $g$. The task loss is defined as $C^t(x + \delta + h(\phi, x + \delta)) := C(g(x + \delta + h(\phi, x + \delta)), t)$, where $C$ denotes the cross-entropy loss. We then minimize the expected loss over the distribution $\mathcal{B}$: $\min_\delta \mathbb{E}_{\phi \sim \mathcal{B}} \left[ C^t(x + \delta + h(\phi, x + \delta)) \right]$, which can be optimized via gradient descent.*

Let $h(\phi, x + \delta)$ complies with distribution $\mathcal{P}$. By the Law of the Unconscious Statistician (LOTUS), we have: $\mathbb{E}_{\phi \sim \mathcal{B}} \left[ C^t(x + \delta + h(\phi, x + \delta)) \right] = \mathbb{E}_{\eta \sim \mathcal{P}} \left[ C^t(x + \delta + \eta) \right]$, which allows us to formulate

the problem of enhancing robustness against various disturbances as the design of a suitable mapping function $h(\phi, x + \delta)$. Unlike conventional methods that sample $\eta$ from a fixed distribution, function $h(\phi, x + \delta)$ can be designed to adapt both the adversarial example and surrogate models, enabling it to produce the most destructive disturbances for each specific sample. In the following, we analyze three key challenges in applying the proposed robust attack framework:

▷ **Limited Sampling Coverage:** The estimation of the expected loss typically relies on a limited number of Monte Carlo samples. This can lead to poor coverage of the disturbance space, resulting in adversarial examples that generalize poorly to unseen disturbances;

▷ **Distribution Mismatch:** During application, adversarial examples may encounter real-world disturbance that differs significantly from the distribution of $h(\phi, x + \delta)$, causing the attack to fail;

▷ **Transferability Consideration:** Under black-box settings, we also need to account for the transferability of adversarial examples to ensure their effectiveness on the unseen target models.

## 3.2 INVERSE GRADIENT SAMPLING

In this section, we first introduce the **I**nverse **G**radient **S**ampling (IGS) method and then theoretically analyze how it addresses the first limitation of existing approaches, namely the issue of *limited sampling coverage*, as discussed in section 3.1.

Based on the proposed robust attack framework, we define the map function $h(\phi, x + \delta)$ as $h(\phi, x + \delta) = \phi + \nabla_\phi C^t(x + \delta + \phi)$. The **Step 2** is then solved using a two-step iterative approach:

$$h(\phi_j, x + \delta) = \phi_j + \nabla_{\phi_j} C^t(x + \delta + \phi_j), \quad \phi_j \sim \mathcal{B} \tag{1}$$

$$\delta_{i+1} = \delta_i - \alpha \cdot \nabla_\delta \left( \frac{1}{N} \sum_{j=1}^{N} C^t(x + \delta_i + h(\phi_j, x + \delta)) \right). \tag{2}$$

The challenge of *limited sampling coverage* arises from an insufficient number of training samples, such that realistic perturbations may deviate substantially from any learned disturbance $h(\phi_i, x + \delta)$. As a result, adversarial examples may fail to remain effective under real-world disturbance. This suggests that robustness fundamentally depends on whether the set of trained disturbances $\{h(\phi_i, x + \delta)\}$ can sufficiently approximate the most destructive disturbances applied to $x + \delta$.

To quantitatively evaluate the performance of IGS, we assume that the most destructive disturbance is given by $\phi^* = \arg\max_{\|\phi\|<r} C^t(x + \delta + \phi)$, and that $\phi^*$ is uniformly distributed within the neighborhood, i.e., $\phi^* \sim \mathcal{B}(0, r)$. Under this assumption, the average loss over sampled disturbances satisfies $\frac{1}{N} \sum_{i=1}^{N} C^t(x + \delta + h(\phi_i, x + \delta)) \leq C^t(x + \delta + \phi^*)$. We define the error as the gap between the upper bound and the empirical average: $\mathcal{E} := C^t(x + \delta + \phi^*) - \frac{1}{N} \sum_{i=1}^{N} C^t(x + \delta + h(\phi_i, x + \delta))$. During the iterative optimization in Equation (2), the accumulated error in $\delta$ is given by:

$$\Delta\delta = \sum_I \left[ \nabla_\delta C^t(x + \delta + \phi^*) - \nabla_\delta \left( \frac{1}{N} \sum_{j=1}^{N} C^t(x + \delta + h(\phi_i, x + \delta)) \right) \right] \approx \sum_I \nabla_\delta \mathbb{E}_{\phi \sim \mathcal{B}(0,r)}[\mathcal{E}], \tag{3}$$

where $I$ is the iteration number of Equation 2. Assuming that $C^t$ is Lipschitz continuous in the neighborhood of $x + \delta$, the expected error over the sampling process satisfies $\mathbb{E}_{\phi \sim \mathcal{B}(0,r)}[\mathcal{E}] \leq \mathbb{E}_{\phi \sim \mathcal{B}(0,r)}\|h(\phi, x + \delta) - \phi^*\|$. This enables us to compare different sampling strategies by their ability to minimize $\mathbb{E}_{\phi \sim \mathcal{B}(0,r)}\|h(\phi, x + \delta) - \phi^*\|$. Since $\phi^*$ can appear anywhere within the neighborhood of $x + \delta$, we derive the following theorem to compute the expectation when $h(\phi, x + \delta) = \phi$, which is commonly used in EOT-based approaches Athalye et al. (2018); Hu et al. (2021); Liu et al. (2022).

**Theorem 1** *Let $m$ denotes the dimensionality of the input space, and let $n$ be the number of samples drawn from $\mathcal{B}(0, r)$. Then, $\mathbb{E}_{\phi^* \sim \mathcal{B}(0,r)} \left[ \mathbb{E}_{\phi \sim \mathcal{B}(0,r)} \left[ \|\phi - \phi^*\| \,\middle|\, \phi^* \right] \right] = r \cdot \Gamma\left(\frac{1}{m}\right) \cdot n^{-\frac{1}{m}}$.*

**Remark 1** *Theorem 1 indicates that the expected error decreases with the number of queries $n$ following a power-law decay of $-1/m$. To halve the error, the number of queries must increase by a factor of $2^m$, which becomes computationally prohibitive in high-dimensional spaces.*

To further quantify the advantage of IGS over EOT, we present the following theorem:

**Theorem 2** *Let $C^t$ be a convex function in a spherical neighborhood of radius $r$ centered at $x + \delta$, with a unique extremum point $x + \delta + \phi^*$. Then, the following relation holds: $h(\phi) - \phi = \gamma(\phi^* - \phi)$, where the scalar coefficient $\gamma$ is given by $\gamma = \frac{\|\nabla_\phi C^t(x + \delta + \phi)\|}{\|\phi^* - \phi\|}$.*

Let $E_{\text{IGS}}(\mathcal{E})$ and $E_{\text{EOT}}(\mathcal{E})$ denote the error bounds of IGS and EOT. Based on Theorem 2, we have:

$$E_{\text{IGS}}(\mathcal{E}) = \mathbb{E}_{\phi \sim \mathcal{B}(0,r)} \|h(\phi, x + \delta) - \phi^*\| = (1 - \gamma) \cdot r \cdot \Gamma\left(\frac{1}{m}\right) \cdot n^{-\frac{1}{m}}, \tag{4}$$

$$E_{\text{EOT}}(\mathcal{E}) = \mathbb{E}_{\phi \sim \mathcal{B}(0,r)} \|\phi - \phi^*\| = r \cdot \Gamma\left(\frac{1}{m}\right) \cdot n^{-\frac{1}{m}}. \tag{5}$$

Let $n_{\text{IGS}}$ and $n_{\text{EOT}}$ denote the number of queries required by IGS and EOT, respectively, to achieve the same error bound. By equating the two bounds, we obtain:

$$1 = \frac{1}{1 - \gamma} \cdot \left(\frac{n_{\text{EOT}}}{n_{\text{IGS}}}\right)^{-\frac{1}{m}} \Rightarrow \frac{n_{\text{EOT}}}{n_{\text{IGS}}} = \frac{1}{(1 - \gamma)^m}. \tag{6}$$

**Remark 2** *Equation (6) shows that the efficiency advantage of IGS over EOT scales exponentially with the data dimensionality $m$. In typical vision tasks where $m > 10^4$, this advantage becomes particularly pronounced. For example, on ImageNet ($m$=256×256×3) with $\gamma \approx 10^{-4}$, we estimate:*

$$\frac{n_{IGS}}{n_{IGS}} \approx 3.5 \times 10^8.$$

*This demonstrates that IGS can capture the most destructive disturbance using far fewer samples than EOT, significantly alleviating the issue of limited sampling coverage. Theoretical extension for non-convex conditions are detailed in Appendix C.*

### 3.3 ROBUSTNESS AND TRANSFERABILITY UNDER DISTRIBUTIONS MISMATCH

Adversarial examples often encounter disturbances that deviate from the distribution assumed during training. In this section, we analyze why our method maintains strong performance under such *distribution mismatch* and examine how it promotes the *transferability* of adversarial examples.

#### 3.3.1 WHAT DETERMINES THE ROBUSTNESS OF ADVERSARIAL EXAMPLES?

Our analysis is motivated by the observation that clean samples exhibit significantly greater robustness under disturbance compared to existing adversarial examples. To quantify this, we define the *robustness boundary*, denoted as $\mathcal{K}_S^\tau$, as the minimum amount of disturbance required to change the model's prediction on a given sample set $S$. Formally: $\mathcal{K}_S^\tau = \| \arg\max_\theta [\mathbb{E}_{x \in S}[Z_\theta] < \tau] \|$, $Z_\theta = g(x + \theta) \cdot \mathbf{1}_{\{i=\text{top}k\}}(g(x))$, where $\theta$ is random disturbance, $\tau$ is a confidence threshold, and $\mathbf{1}_{\{i=\text{top}k\}}(\cdot)$ indicates whether the prediction belongs to the original top-$k$ classes. The robustness boundary $\mathcal{K}_S^\tau$ indicates the ability of samples in $S$ to retain their original labels under disturbance.

As shown in Fig. 2, the robustness bounds $\mathcal{K}_{S_{\text{ori}}}^\tau$ are consistently larger for clean samples than for adversarial examples, indicating that clean samples are more robust to disturbance. This motivates the conjecture that *higher likelihood under the natural data distribution $P_D$ correlates with greater robustness.* Clean samples are drawn from $P_D$, while adversarial examples are typically deviate from $P_D$ Zhu et al. (2022). Since models are trained to fit $P_D$, they tend to generalize better to samples that are more likely under $P_D$, which explains the superior robustness of clean samples. However, directly computing $P_D(x_{\text{adv}})$ is generally intractable. The key question, therefore, becomes: *How can we construct adversarial samples that maintain a high likelihood under $P_D$?* We address this question in the following section.

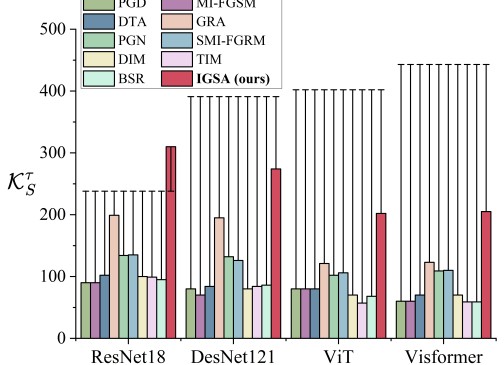

Figure 2: A comparison of the robustness bounds $\mathcal{K}_S^\tau$ across different adversarial examples and clean samples. The black bars represent the robustness bounds for clean samples.

#### 3.3.2 ALIGNING GRADIENTS FOR HIGH-LIKELIHOOD ADVERSARIAL SAMPLES

In Equation 2, we perform gradient descent on $C^t$ to push the input toward misclassification. This process increases the surrogate model's confidence in the target class, $g^t(x + \delta)$. When the gradient

---

**Algorithm 1** Inverse Gradient Sample Adversarial Attack

---

**Input:** Original data $x$; balancing parameter $\lambda$; inverse learning rate $\mu$; step size $\alpha$; perturbation preset range $\epsilon$; number of sampling points $N$; sampling variance $\sigma^2$; model $g$; loss function $C$; sign function $\text{sign}(\cdot)$; target class $t$.
**Output:** Robust adversarial example $x_{adv}$.
Initialize adversarial perturbation $\delta \leftarrow 0$ and cumulative update direction $\mathbf{d}_{\text{sum}} \leftarrow 0$.
**repeat**
    $x_{adv} \leftarrow x + \delta$
    **for** $i = 1$ to $N$ **do**
        Sample $\phi_i \sim \mathcal{N}(0, \sigma^2)$
        Compute loss at $\phi_i$:              $C_{\phi_i}^t = C(g(x + \delta + \phi_i), t) + \lambda \cdot |\delta|$
        Gradient update for $\phi_i$:          $\hat{\phi}_i \leftarrow \phi_i + \mu \cdot \text{sign}(\nabla_\phi C_{\phi_i}^t)$
        Compute loss at $\hat{\phi}_i$:            $C_{\hat{\phi}_i}^t = C(g(x + \delta + \hat{\phi}_i), t) + \lambda \cdot |\delta|$
        Compute update direction:       $\mathbf{d}_{\hat{\phi}_i} \leftarrow C_{\hat{\phi}_i}^t \cdot \nabla_{\phi_i} \log \mathcal{N}(x + \delta + \phi_i; x + \delta, \sigma^2)$
        Accumulate $\mathbf{d}_{sum}$:              $\mathbf{d}_{sum} \leftarrow \mathbf{d}_{sum} + \mathbf{d}_{\hat{\phi}_i}$.
    **end for**
    Update adversarial perturbation:    $x_{\text{adv}} = x_{\text{adv}} - \alpha \cdot \text{sign}(\mathbf{d}_{\text{sum}})$
    Constrain perturbation magnitude:  $\delta \leftarrow clamp[-\epsilon, \epsilon]$
**until** Loss $C_{\phi_i}^t$ converges
$x_{adv} \leftarrow x + \delta$
**return** $x_{adv}$

---

of the surrogate model aligns with $P_D$, the update also increases the likelihood of adversarial example under $P_D$. To encourage the search for a perturbation $\delta$ where such alignment occurs between the gradients of the surrogate model and $P_D$, we present Theorem 3.

**Theorem 3** *Let $tr(H[\cdot])$ denote the trace of the Hessian matrix, and let $\mathcal{B}(0, r)$ represent a uniform distribution over the ball of radius $r$ in $\mathbb{R}^m$. Then:*

$$\nabla_\delta \mathbb{E}_{\mathcal{B}(0,r)}\left[(\nabla_\delta C^t)^T \cdot \nabla_\delta P_D\right] = -\nabla_\delta \mathbb{E}_{\mathcal{B}(0,r)}^{P_D}\left[tr(H[C^t])\right].$$

**Remark 3** *Theorem 3 establishes that minimizing the trace of the Hessian of $C^t$ enhances the alignment between $\nabla_\delta C^t$ and $\nabla_\delta P_D$, which in turn increases the likelihood of the resulting adversarial examples under $P_D$ during the iterative process.*

We now examine whether the iterative optimization procedure implemented by our proposed IGS method leads to adversarial examples with reduced Hessian trace.

**Theorem 4** *Let $C^t$ denote $C^t(x + \delta)$. Suppose the Hessian $H[C^t]$ is bounded in the neighborhood of $x + \delta$, such that $\|H[C^t]\|_2 \leq L$, the update rule satisfies:*

$$\nabla_\delta \mathbb{E}_\phi\left[C^t(x + \delta + \phi + \nabla_\phi C^t)\right] = \nabla_\delta C^t + \|\nabla_\delta C^t\|^2 + \frac{\sigma^2}{2} \nabla_\delta tr(H[C^t]) + \mathcal{O}(\sigma^4), \sigma^2 \ll 1/L$$

**Remark 4** *Theorem 4 shows that the proposed IGS method implicitly minimizes $tr(H[C^t])$ throughout the iterative process. This contributes to an increased likelihood of adversarial examples under the data distribution $P_D$, which is verified by Fig. 3. At the same time, IGS also reduces $\|\nabla_\delta C^t\|^2$, promoting smoother loss landscapes. As demonstrated by Ge et al. (2023), such improvements in smoothness significantly enhance the transferability of adversarial examples across models.*

### 3.4 INVERSE GRADIENT SAMPLE ADVERSARIAL ATTACK

We present the detailed implementation of the IGSA in Algorithm 1. In Algorithm 1, we incorporate several practical techniques to improve convergence and efficiency:

**(1) Sampling Distribution.** We sample the disturbance $\phi$ from a Gaussian distribution, $\phi \sim \mathcal{N}(0, \sigma^2)$ leads to faster convergence and more stable optimization.

**(2) Efficient Gradient Estimation.** To reduce the computational cost associated with second-order derivatives in Equations equation 3 and equation 2, we propose an efficient first-order approximation based on Theorem 5 in appendix:

$$\begin{aligned}
&\nabla_\delta \mathbb{E}_\phi\left[C^t(x + \delta + \phi + \nabla_\phi C^t)\right] \\
&\approx \mathbb{E}_{\phi \sim \mathcal{N}(0,\sigma^2)}\left[C^t(x + \delta + \phi + \nabla_\phi C^t) \cdot \nabla_\delta \log \mathcal{N}(x + \delta + \phi; x + \delta, \sigma^2)\right].
\end{aligned} \tag{7}$$

Table 1: Robustness of various attacks on ImageNet under *additive and non-additive* disturbances.

| ASR (%) | VGG19 | | | | ResNet34 | | | | ViT | | | | Avg. time |
|---|---|---|---|---|---|---|---|---|---|---|---|---|---|
| | Additive | | Non-additive | | Additive | | Non-additive | | Additive | | Non-additive | | |
| Disturbance Types → 
 Attacks Types ↓ | GSB | JPEG | RT | CB | GSB | JPEG | RT | CB | GSB | JPEG | RT | CB | |
| PGD Madry et al. (2017) | 8.3 | 2.1 | 43.8 | 0.0 | 0.3 | 31.3 | 4.2 | 0.0 | 9.3 | 17.2 | 6.1 | 0.0 | 0.025 |
| MI-FGSM Dong et al. (2018) | 66.7 | 62.5 | 72.9 | 0.0 | 77.1 | 87.5 | 16.7 | 0.0 | 67.4 | 79.6 | 30.9 | 2.0 | 0.025 |
| DTA Yang et al. (2023) | 70.8 | 68.8 | 75.0 | 2.1 | 91.7 | **100.0** | 18.8 | 0.0 | 78.1 | 78.1 | 42.3 | 7.5 | 0.186 |
| GRA Zhu et al. (2023a) | 62.5 | 64.6 | 52.1 | 8.3 | 89.6 | 93.8 | 56.3 | 12.5 | 81.9 | 85.4 | 76.4 | 19.5 | 0.345 |
| PGN Ge et al. (2023) | 33.3 | 41.7 | 27.1 | 8.3 | 72.9 | 81.3 | 37.5 | 8.3 | 72.6 | 70.9 | 65.8 | 11.7 | 0.659 |
| SMI-FGRM Han et al. (2023) | 66.7 | 62.5 | 52.1 | 12.5 | 87.5 | 93.8 | 39.6 | 6.3 | 84.0 | 86.7 | 65.0 | 13.9 | 0.198 |
| DIM Xie et al. (2019) | **87.5** | 75.0 | 66.7 | 29.2 | 91.7 | 93.8 | 39.6 | 12.5 | 89.3 | 82.3 | 79.6 | 20.4 | 0.020 |
| TIM Dong et al. (2019) | 68.8 | 58.3 | 27.1 | 12.5 | 87.5 | 93.8 | 8.3 | 4.2 | 81.8 | 82.8 | 19.2 | 3.9 | 0.020 |
| BSR Wang et al. (2024a) | 39.6 | 31.3 | 83.3 | 10.4 | 68.8 | 68.8 | 75.0 | 8.3 | 71.7 | 69.9 | 82.7 | 11.6 | 0.203 |
| PGD+EOT Athalye et al. (2018) | **87.5** | 93.8 | 79.2 | 27.1 | 91.7 | **100.0** | 40.3 | 22.9 | 87.6 | 89.2 | 70.9 | 26.7 | 0.461 |
| **IGSA (ours)** | **87.5** | **95.8** | **96.7** | **35.4** | **97.2** | **100.0** | **75.0** | **50.8** | **92.2** | **91.9** | **85.3** | **27.0** | 0.423 |

**(3) Gradient Magnitude Control.** To ensure stable optimization, we employ a sign-based gradient update rule. Additionally, an $\ell_2$-norm constraint is imposed on $\delta$ during the computation of the loss $C^t$, in order to minimize the magnitude of the perturbation introduced.

# 4 EXPERIMENTS

## 4.1 EXPERIMENT SETUP

**Tasks and Models.** We evaluate our proposed IGSA on two types of tasks, including image classification and face recognition. We use two benchmark datasets in the image classification task, including CIFAR-10 Krizhevsky et al. (2009) and ImageNet Deng et al. (2009). The models on the ImageNet dataset use the official pre-trained models from torchvision, including VGG19 Simonyan (2014), ResNet34 He et al. (2016), ResNet101 He et al. (2016), ViT-Base Dosovitskiy et al. (2020), and Inception-v3 Szegedy et al. (2016). The models on the CIFAR-10 dataset, including VGG19, ResNet34, and ViT-Base, are trained using the standard cross-entropy loss. In the face recognition task, we use the CelebA dataset Liu et al. (2015) and perform attacks based on the aggregation models of the insightface framework Deng et al. (2019). We train these models using a pairwise loss function. More implementation is detailed in Appendix B.1.

**Attack Settings.** In our experiments, the attack success rate (ASR) is mainly used to measure the performance of various attacks. All experiments are conducted using a NVIDIA 4090 GPU. In various tasks, we first set the adversarial perturbation strength $\epsilon$ of various attacks to a unified value to ensure fairness in comparison. In experiments on the CIFAR-10 and CelebA datasets, $\epsilon$ is set to $8/255$; in experiments on the ImageNet dataset, $\epsilon$ is set to $16/255$. The number of iterations of various attacks is uniformly set to 100. For our proposed IGSA, the hyperparameter $\alpha$ for adversarial perturbation update is set to $1.6/255$. The number of sampling points $N$ is set to 20. The hyperparameter $\lambda$ is set to 0.1 on the ImageNet dataset and 0.03 on the CIFAR-10 dataset. The hyperparameter $\mu$ is set to 0.4 for the ImageNet dataset. For the CIFAR-10 dataset, $\mu$ varies across different models: 0.5 for ResNet, 0.8 for VGG, and 0.3 for ViT.

## 4.2 ROBUSTNESS EXPERIMENTS

**Evaluation of Attack Robustness:** To evaluate the robustness of various attack methods, we conduct targeted attacks on the ImageNet dataset under both additive and non-additive disturbances (Table 1). For additive disturbances, we apply Gaussian blur (GSB) with a kernel size of 5 and standard deviation 1.0, and JPEG compression at $50\%$. For non-additive disturbances, we use rotation transformation (RT) with a $10°$ angle and combined transformation (CB), including resizing ($\times 1.15$), rotation ($5°$), and perspective distortion (0.15). Additional experiments are detailed in the Appendix B.2.1.

**IGSA Performance Under Disturbances:** Without disturbances, most attacks achieve nearly $100\%$ ASR. Table 1 shows IGSA outperforms existing methods under almost all disturbances. Transfer-based attacks like MI-FGSM Dong et al. (2018), DTA Yang et al. (2023), and GRA Zhu et al. (2023a) show significant performance drops under disturbances. Robustness-oriented attacks such as DIM Xie et al. (2019), TIM Dong et al. (2019), BSR Wang et al. (2024a), and EOT Athalye et al. (2018) generate more robust adversarial examples, but their effectiveness is limited when the enhancement strategy mismatches the actual disturbance.

Table 2: Robustness of various attacks on the ImageNet dataset against defended models.

| ASR (%) | ResNet50 (Defense) | | ViT (Defense) | |
|---|---|---|---|---|
| Attack Types | un-tar | tar | un-tar | tar |
| MI-FGSM Dong et al. (2018) | 87.12 | 11.90 | 71.95 | 5.60 |
| DTA Yang et al. (2023) | 69.72 | 4.60 | 81.89 | 11.20 |
| GRA Zhu et al. (2023a) | 94.69 | 16.10 | 72.45 | 4.90 |
| PGN Ge et al. (2023) | **96.38** | 14.60 | 69.56 | 4.00 |
| SMI-FGRM Han et al. (2023) | 86.89 | 15.30 | 73.84 | 5.80 |
| DIM Xie et al. (2019) | 90.40 | 13.60 | 72.08 | 4.30 |
| TIM Dong et al. (2019) | 94.01 | 18.60 | 62.52 | 2.90 |
| BSR Wang et al. (2024a) | 91.64 | 18.20 | 68.43 | 2.90 |
| **IGSA (ours)** | 91.75 | **27.30** | **90.94** | **23.90** |

Table 3: Black-box testing of various attacks on the ImageNet dataset.

| ASR (%) | ResNet | ViT |
|---|---|---|
| Attacks Types ↓ | | |
| DIM Xie et al. (2019) | 78.0 | 66.4 |
| DTA Yang et al. (2023) | 66.7 | 61.7 |
| SMI-FGRM Han et al. (2023) | 70.8 | 52.0 |
| ILPD Li et al. (2024) | 86.0 | 83.1 |
| **IGSA (ours)** | 83.0 | 79.2 |
| DIM Xie et al. (2019) + **IGSA** | 91.0 | **89.6** |
| DTA Yang et al. (2023) + **IGSA** | 83.0 | 78.5 |
| SMI-FGRM Han et al. (2023) + **IGSA** | **91.7** | 77.4 |
| ILPD Li et al. (2024) + **IGSA** | 89.0 | 82.7 |

Table 4: Black-box testing of various attacks on the face recognition task using the CelebA dataset.

| ASR (%) | ResNet50 | | | | MBF | | | |
|---|---|---|---|---|---|---|---|---|
| Disturbance Types → Attack Types ↓ | RS | RT | PT | CB | GSB | CTRS | BRT | JPEG |
| MI-FGSM Dong et al. (2018) | 66.7 | 80.0 | 82.2 | 84.4 | 27.3 | 38.6 | 50.0 | 43.2 |
| DTA Yang et al. (2023) | 79.5 | 90.9 | 88.6 | 87.8 | 43.9 | 58.5 | 63.4 | 63.4 |
| GRA Zhu et al. (2023a) | 85.4 | 73.2 | 87.8 | 90.2 | 61.0 | 47.6 | 54.8 | 61.9 |
| PGN Ge et al. (2023) | 85.4 | 92.1 | 92.7 | 92.7 | 56.8 | 40.5 | 48.6 | 59.5 |
| SMI-FGRM Han et al. (2023) | 55.3 | 48.9 | 61.7 | 57.4 | 25.5 | 23.4 | 40.4 | 36.2 |
| DIM Xie et al. (2019) | 86.0 | 83.7 | 83.7 | 86.0 | 58.1 | 51.2 | 58.1 | 62.8 |
| TIM Dong et al. (2019) | 63.8 | 10.6 | 23.4 | 23.4 | 22.9 | 4.2 | 6.3 | 4.2 |
| **IGSA (ours)** | **87.2** | **92.3** | **92.7** | **94.9** | **61.0** | **68.3** | **73.2** | **70.7** |

**Further Validation on CIFAR-10:** We validate IGSA on CIFAR-10; detailed results are in Appendix B.2.2. Recent works focus on generating adversarial examples resilient to physical-world distortions like reshooting, rotation, and lighting changes. We benchmark IGSA against state-of-the-art physical-world attack methods, also detailed in the Appendix B.2.2.

**Performance Against Defended Models:** We evaluate IGSA against defended models, including ResNet50 and ViT models trained using defense method of adversarial training Liu et al. (2025) within the ARES 2.0 framework Dong et al. (2020) (Table 2). Results show IGSA maintains significantly higher robustness than existing attacks, especially under targeted settings where other approaches largely fail.

### 4.3 TRANSFERABILITY EXPERIMENTS

**Evaluation Setup for Transferability:** We evaluate the transferability of various attacks on image classification (ImageNet) and face recognition (CelebA), as shown in Table 3 and Table 4. Inception-v3 is used as the surrogate model. For black-box evaluation, adversarial examples are tested on ResNet34 and ViT-base for image classification, and ResNet50 and MBF for face recognition. We apply IGSA to enhance state-of-the-art transfer attack methods in image classification and test robustness under black-box settings in face recognition. Additional implementation details are provided in Appendix B.1.

**Transfer Performance of IGSA:** Experimental results show that IGSA achieves higher ASR than existing transfer attacks. When applied to DIM Xie et al. (2019), DTA Yang et al. (2023), SMI-FGRM Han et al. (2023), and ILPD Li et al. (2024), the ASR against ResNet34 increases by 13.0%, 16.3%, 20.9%, and 3.0%, respectively; against ViT, improvements are 19.0%, 12.4%, 22.9%, and 3.0%. Under black-box robustness tests, IGSA consistently outperforms all baseline attacks across disturbance types.

### 4.4 HYPERPARAMETER ANALYSIS AND ABLATION STUDY

**Hyperparameter analysis.** We conducted a hyperparameter analysis of IGSA using the ResNet101 model. The results are shown by Table 5. It can be seen that IGSA achieves an ASR above 90% when the number of iterations exceeds 50. The parameter $\lambda$ has a negative impact on ASR, as it

constrains the magnitude of perturbations during iterations. The number of sampling points, $t_{\text{num}}$, significantly boosts ASR from 94.4% at 5 points to 100% at 25 points by providing richer neighborhood information. The learning rate, $\alpha$, exhibits an optimal range, with ASR peaking at 99% for $\alpha = 1.6/255$ and slightly decreasing for higher values. The parameter $\mu$ has a relatively minor effect on ASR, varying from 96.0% to 98.7%.

**Ablation analysis of IGS.** As discussed in Section 3.2, IGS enhances adversarial example robustness by approximating the most disruptive disturbance. To evaluate its impact, we compare IGSA with various EOT-based attacks. Figure 2 shows that under strong disturbance (SNR=10), IGSA achieves over 80% attack success rate with only 5 sampling iterations, while EOT-based attacks reach only about 60% with 50 samples. This demonstrates that IGS significantly improves both efficiency and effectiveness in generating robust adversarial examples.

**Likelihood Analysis.** As discussed in Section 3.3, our IGS enhances robustness by aligning the

Table 5: Hyperparameter analysis of our proposed IGSA, where the shaded values are used in comparative experiment.

| $\lambda$ | 0.02 | 0.05 | 0.10 | 0.20 | 0.30 |
|---|---|---|---|---|---|
| ASR (%) | 100.00 | 100.00 | 97.22 | 69.44 | 5.56 |
| $\mu$ | 0.02 | 0.04 | 0.10 | 0.20 | 0.50 |
| ASR (%) | 96.00 | 97.22 | 97.22 | 97.37 | 98.68 |
| iteration | 10 | 20 | 50 | 100 | 200 |
| ASR (%) | 11.11 | 56.00 | 94.44 | 97.22 | 100.00 |
| $N$ | 5 | 10 | 15 | 20 | 25 |
| ASR (%) | 94.44 | 97.22 | 97.22 | 99.00 | 100.00 |
| $\alpha$ | 0.4/255 | 0.8/255 | 1/255 | 1.6/255 | 3.2/255 |
| ASR (%) | 88.00 | 96.00 | 97.22 | 99.00 | 97.22 |

likelihood of adversarial examples with the original data distribution. We validated this using an energy-based out-of-distribution detection method Liu et al. (2020b). Results show that IGSA-generated samples achieve in-distribution scores comparable to clean data, whereas those from other attacks show significantly lower scores. This demonstrates that IGSA produces more realistic and distribution-aware adversarial examples, improving both robustness and stealth.

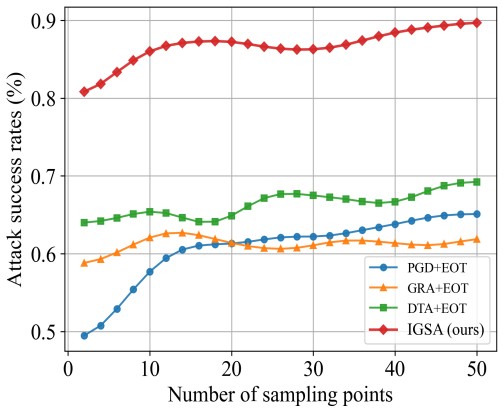

Figure 4: Comparison of attack success rates between IGSA and EOT as the number of samples increases.

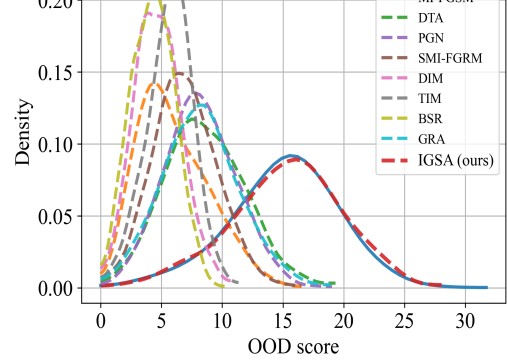

Figure 3: Distribution of the Energy OOD scores Liu et al. (2020b) for the clean samples, CIFAR-10, and the adversarial examples.

## 5 CONCLUSION AND LIMITATION DISCUSSION

In this paper, we propose a robust adversarial attack framework to address the vulnerability of transfer-based attacks under various disturbances. Within this framework, we introduce IGSA to tackle three key challenges: sampling coverage limitation, distribution mismatch, and transferability. Extensive experiments show that IGSA significantly outperforms existing methods in robustness against diverse unknown disturbances on both image recognition and face recognition tasks. Moreover, IGSA achieves strong transferability, making it highly effective in black-box settings. One limitation of our current work is the use of a fixed mapping function $h(\phi, x + \delta)$ for disturbance sampling. Replacing it with a learnable module could further enhance the adaptability and effectiveness of the robust attack framework, which we leave for future exploration. We hope our work inspires more research into generating adversarial examples that are both transferable and robust under real-world variations.

## 6 ACKNOWLEDGEMENTS

This work is supported by the National Defense Basic Scientific Research Program of China (No. JCKY2023603C043), the Key RD Plan of Heilongjiang Province (No. 2022ZX01C01), Natural Science Foundation of Heilongjiang Province of China (No. LH2024F023), the National Natural Science Foundation of China under Grant 62472128, and Grant 62172402.

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

## A    PROOF OF THEOREMS

**Theorem 1** *Let $m$ denote the dimensionality of the input space, and let $n$ be the number of samples drawn from $\mathcal{B}(0, r)$. Then, $\mathbb{E}_{\phi^* \sim \mathcal{B}(0,r)} \left[ \mathbb{E}_{\phi \sim \mathcal{B}(0,r)} \left[ \|\phi - \phi^*\| \,\big|\, \phi^* \right] \right] = r \cdot \Gamma\left(\frac{1}{m}\right) \cdot n^{-\frac{1}{m}}$.*

**Proof 1** We define the random variable $\|h(\phi_i) - \phi^*\|$ as $Z_i$, and the random variable $\min\{Z_1, Z_2, ... Z_N\}$ as $Y$. We know that the volume of an $m$-dimensional hypersphere is given by $V_m(r) = \frac{\pi^{m/2}}{\Gamma(m/2+1)} \cdot r^m = A \cdot r^m$, where $A = \frac{\pi^{m/2}}{\Gamma(m/2+1)}$. Since $\phi \sim \mathcal{B}(0, r)$, the probability density function is $f(\phi) = 1/V_m(r)$. For $Z_i$, its cumulative distribution function is $F_{Z_i}(z) = V_m(z)/V_m(r)$. The probability density function $f_{Z_i}(z)$ is:

$$f_{Z_i}(z) = (F_{Z_i}(z))'_z = (\frac{z^m}{r^m})'_z = \frac{m \cdot z^{m-1}}{r^m} \tag{8}$$

Thus, the cumulative distribution function of $Y$ can be computed as:

$$\begin{aligned} F_Y(y) &= 1 - P(Z_1 > y, Z_2 > y, ..., Z_N > y) \\ &= 1 - \prod_{i=1}^{N} P(Z_i > y) = 1 - \prod_{i=1}^{N}(1 - F_{Z_i}(y)) \\ &= 1 - (1 - F_{Z_i}(y))^n \end{aligned} \tag{9}$$

Equation (9) holds because $Z_1, Z_2, ... Z_N$ are independent. Therefore, we can compute the probability density function of $Y$ as:

$$\begin{aligned} f_Y(y) &= F'_Y(y) = n(1 - F_{Z_i}(y))^{n-1} \cdot f_{Z_i}(y) \\ &= n(1 - (\frac{y}{r})^m)^{n-1} \cdot m \cdot \frac{y^{m-1}}{r^m} \end{aligned} \tag{10}$$

Now, we can compute the inner expectation in Theorem 1 as:

$$\begin{aligned} \mathbb{E}_{\phi \sim \mathcal{B}(0,r)} \left[ \|h(\phi) - \phi^*\| \big| \phi^* \right] &= \int_0^r y \cdot f_Y(y) dy \\ &= \int n(1 - (\frac{y}{r})^m)^{n-1} \cdot m \cdot (\frac{y}{r})^m dy \end{aligned} \tag{11}$$

Let $u = (\frac{y}{r})^m$, where $0 < u < 1$, then we have:

$$\begin{aligned} \mathbb{E}_{\phi \sim \mathcal{B}(0,r)} \left[ \|h(\phi) - \phi^*\| \big| \phi^* \right] &= \int_0^1 n(1-u)^{n-1} \cdot\cdot u d(r \cdot u^{\frac{1}{m}}) \\ &= r \cdot n \int_0^1 (1-u)^{n-1} \cdot u \cdot u^{\frac{1}{m}-1} du \\ &= r \cdot n \int_0^1 (1-u)^{n-1} u^{\frac{1}{m}} du \\ &\stackrel{(1)}{=} r \cdot n \cdot \frac{\Gamma(n)\Gamma(\frac{1}{m}+1)}{\Gamma(n+\frac{1}{m}+1)} \end{aligned} \tag{12}$$

By the Beta function: $\beta(a, b) = \int_0^1 (1-x)^{a-1} \cdot x^{b-1} = \frac{\Gamma(a)\Gamma(b)}{\Gamma(a+b)}$, the equality $\stackrel{(1)}{=}$ holds.

Next, we consider the case when the sample size $n$ is large. By Stirling's approximation Feller (1967): $\Gamma(x) \approx \sqrt{2\pi x} \cdot (\frac{x}{e})^x$, we express $\Gamma(n)$ and $\Gamma(n + \frac{1}{m} + 1)$ in equation (12) as:

$$\begin{aligned} \Gamma(n) &= \sqrt{2\pi n} \cdot (\frac{n}{e})^n \\ \Gamma(n + \frac{1}{m} + 1) &= \sqrt{2\pi(n + \frac{1}{m} + 1)} \cdot \left(\frac{n + \frac{1}{m} + 1}{e}\right)^{n+\frac{1}{m}+1} \end{aligned} \tag{13}$$

By the Gamma function: $\Gamma(\frac{1}{m}+1) = (\frac{1}{m}+1) \cdot \Gamma(\frac{1}{m})$, equation (12) is transformed into:

$$
\begin{aligned}
\mathbb{E}_{\phi \sim \mathcal{B}(0,r)} \left[ \|h(\phi) - \phi^*\| \Big| \phi^* \right] &= r \cdot n \cdot \frac{\Gamma(n)\Gamma(\frac{1}{m}+1)}{\Gamma(n+\frac{1}{m}+1)} \\
&= \frac{r \cdot n\sqrt{2\pi n} \cdot (\frac{n}{e})^n \cdot (\frac{1}{m}+1) \cdot \Gamma(\frac{1}{m}) \cdot (\frac{1}{m}+1) \cdot \Gamma(\frac{1}{m})}{\sqrt{2\pi(n+\frac{1}{m}+1)} \cdot (\frac{n+\frac{1}{m}+1}{e})^{n+\frac{1}{m}+1}} \\
&= e^{\frac{1}{m}+1} \cdot r \cdot n \cdot \sqrt{\frac{n}{n+\frac{1}{m}+1}} \cdot \frac{n^n}{(n+\frac{1}{m}+1)^{n+\frac{1}{m}+1}} \cdot \Gamma(\frac{1}{m}) \\
&= e^{\frac{1}{m}+1} \cdot r \cdot \Gamma(\frac{1}{m}) \cdot \frac{n^{n+\frac{3}{2}}}{(n+\frac{1}{m}+1)^{n+\frac{1}{m}+\frac{3}{2}}} \\
&= e^{\frac{1}{m}+1} \cdot r \cdot \Gamma(\frac{1}{m}) \cdot n^{-\frac{1}{m}} \cdot \left(\frac{n}{n+\frac{1}{m}+1}\right)^{n+\frac{1}{m}+\frac{3}{2}} \\
&\overset{(2)}{=} r \cdot \Gamma(\frac{1}{m}) \cdot n^{-\frac{1}{m}}
\end{aligned}
\tag{14}
$$

$\overset{(2)}{=}$ holds for when $n$ to $\infty$, $\left(\dfrac{n}{n+\frac{1}{m}+1}\right)^{n+\frac{1}{m}+\frac{2}{3}}$ to $e^{-(\frac{1}{m}+1)}$.

From equation (14), we note that the inner expectation $\mathbb{E}_{\phi \sim \mathcal{B}(0,r)} \left[ \|h(\phi) - \phi^*\| \Big| \phi^* \right]$ is independent of the position of $X^*$. Therefore:

$$
\mathbb{E}_{\phi \sim \mathcal{B}(0,r)}[\mathcal{E}] \leq \mathbb{E}_{\phi \sim \mathcal{B}(0,r)}\|h(\phi) - \phi^*\| = r \cdot \Gamma(\frac{1}{m}) \cdot n^{-\frac{1}{m}}
\tag{15}
$$

**Theorem 2** *Let $C^t$ be a convex function in a spherical neighborhood of radius $r$ centered at $x + \delta$, with a unique extremum point $x + \delta + \phi^*$. Then, the following relation holds: $h(\phi) - \phi = \gamma(\phi^* - \phi)$, where the scalar coefficient $\gamma$ is given by $\gamma = \frac{\|\nabla_\phi C^t(x+\delta+\phi)\|}{\|\phi^* - \phi\|}$.*

**Proof 2** We start from the definition of $h(\phi)$:

$$
h(\phi) - \phi = \nabla_\phi C^t(x + \delta + \phi).
\tag{16}
$$

Since $C^t$ is convex and has a unique extremum (minimum or maximum) at $x + \delta + \phi^*$, we apply the *first-order Taylor expansion* of $C^t$ around $x + \delta + \phi^*$:

$$
C^t(x + \delta + \phi) - C^t(x + \delta + \phi^*) = \nabla_\phi C^t(x + \delta + \xi)^T (\phi - \phi^*),
\tag{17}
$$

for some $\xi$ on the line segment between $\phi$ and $\phi^*$. When $r$ is sufficiently small, we can approximate:

$$
\nabla_\phi C^t(x + \delta + \xi) \approx \nabla_\phi C^t(x + \delta + \phi),
\tag{18}
$$

which gives us:

$$
C^t(x + \delta + \phi) - C^t(x + \delta + \phi^*) \approx \nabla_\phi C^t(x + \delta + \phi)^T (\phi - \phi^*).
\tag{19}
$$

Now consider the first-order Taylor expansion of $C^t$ at $x + \delta + \phi^*$:

$$
C^t(x + \delta + \phi) = C^t(x + \delta + \phi^*) + \nabla_\phi C^t(x + \delta + \phi^*)^T (\phi - \phi^*) + o(\|\phi - \phi^*\|).
\tag{20}
$$

Because $\phi^*$ is an extremum, the gradient at that point vanishes:

$$
\nabla_\phi C^t(x + \delta + \phi^*) = 0.
\tag{21}
$$

Substituting into Equation (20), we obtain:

$$
C^t(x + \delta + \phi) - C^t(x + \delta + \phi^*) = \nabla_\phi C^t(x + \delta + \phi)^T (\phi - \phi^*) + o(\|\phi - \phi^*\|).
\tag{22}
$$

Next, for any direction $d \in \mathbb{R}^d$ with $\|d\| = 1$, and for any small $w > 0$, the local extremality implies:

$$C^t(x + \delta + \phi^*) \geq C^t(x + \delta + wd). \tag{23}$$

Expanding both sides using the Taylor approximation yields:

$$C^t(x + \delta + \phi^*) \geq C^t(x + \delta + \phi) + w\nabla_\phi C^t(x + \delta + \phi)^T d + o(w). \tag{24}$$

Taking $w \to 0$, this inequality must hold for all directions $d$, which implies that $\nabla_\phi C^t(x + \delta + \phi)$ is collinear with $\phi^* - \phi$. That is, there exists a scalar $\gamma' > 0$ such that:

$$\nabla_\phi C^t(x + \delta + \phi) = \gamma'(\phi^* - \phi). \tag{25}$$

Substituting this back into Equation (16), we get:

$$h(\phi) - \phi = \nabla_\phi C^t(x + \delta + \phi) = \gamma(\phi^* - \phi). \tag{26}$$

**Theorem 3** *Let $tr(H[\cdot])$ denote the trace of the Hessian matrix, and let $\mathcal{B}(0, r)$ represent a uniform distribution over the ball of radius $r$ in $\mathbb{R}^m$. Then:*

$$\nabla_\delta \mathbb{E}_{\mathcal{B}(0,r)}[\nabla_\delta(C^t)^T \cdot \nabla_\delta P_D] = -\nabla_\delta \mathbb{E}_{\mathcal{B}(0,r)}^{P_D}[tr(H[C^t])].$$

**Proof 3** Let $V_{\mathcal{B}^m}$ denote the volume of the neighborhood of the sample $\mathcal{B}^m(0, r)$.

$$
\begin{aligned}
\mathbb{E}_{\mathcal{B}^m}[\nabla_\delta(C^t)^T \cdot \nabla_\delta P_D] &= \int_{\mathcal{B}} \frac{1}{V_{\mathcal{B}^m}} \cdot \nabla_\delta(C^t)^T \cdot \nabla_\delta P_D d\delta \\
&= \frac{1}{V_{\mathcal{B}^m}} \cdot \int_{\mathcal{B}^m} \nabla_\delta(C^t)^T \cdot \nabla_\delta P_D d\delta \\
&= \frac{1}{V_{\mathcal{B}^m}} \cdot \sum_{i=1}^m \int_{\mathcal{B}^m} \nabla_{\delta_i} C^t \cdot \nabla_{\delta_i} P_D d\delta \cdot \\
&= \frac{1}{V_{\mathcal{B}^m}} \cdot \sum_{i=1}^m \int_{\mathcal{B}^{m-1}} \left[ \int_a^b \nabla_{\delta_i} C^t \cdot \nabla_{\delta_i} P_D d\delta_i \right] d\delta_{m-1} \\
&\overset{(3)}{=} \frac{1}{V_{\mathcal{B}^m}} \cdot \sum_{i=1}^m \int_{\mathcal{B}^{m-1}} \left[ P_D|_b^a \cdot \nabla_{\delta_i} C^t - \int_a^b P_D \nabla_{\delta_i}^2 C^t d\delta_i \right] d\delta_{m-1} \\
&\overset{(4)}{=} -\frac{1}{V_{\mathcal{B}^m}} \cdot \sum_{i=1}^m \left[ \int_{\mathcal{B}^m} P_D \cdot \nabla_{\delta_i}^2 C^t d\delta_m \right] \\
&= -\frac{1}{V_{\mathcal{B}(0,r)}} \cdot \int_{\mathcal{B}} P_D \cdot tr(H[C^t]) d\delta \\
&= -\mathbb{E}_{\mathcal{B}^m}^{P_D}[tr(H_\delta[C^t])]
\end{aligned}
\tag{27}
$$

The equation $\overset{(3)}{=}$ holds due to the application of integration by parts. In the equation $\overset{(4)}{=}$, $a$ and $b$ represent the upper and lower bounds of the values of the element $\delta_i$ within the neighborhood of $B^m$, respectively. When $B^m$ is sufficiently small, the influence of $\delta_i$ on $P_D$ becomes negligible, i.e., $P_D(a) - P_D(b) \approx 0$. Therefore, the term $P_D|_b^a \cdot \nabla_{\delta_i} C^t \approx 0$.

Taking the gradient of both sides of equation (27) yields:

$$\nabla_\delta \mathbb{E}_{\mathcal{B}(0,r)}[\nabla_\delta(C^t)^T \cdot \nabla_\delta P_D] = -\nabla_\delta \mathbb{E}_{\mathcal{B}(0,r)}^{P_D}[tr(H_\delta[C^t])] \tag{28}$$

**Theorem 4** *Let $C^t$ denote $C^t(x + \delta)$. Suppose the Hessian $H[C^t]$ is bounded in the neighborhood of $x + \delta$, such that $\|H[C^t]\|_2 \leq L$, the update rule satisfies:*

$$\nabla_\delta \mathbb{E}_\phi \left[ C^t(x + \delta + \phi + \nabla_\phi C^t) \right] = \nabla_\delta C^t + \|\nabla_\delta C^t\|^2 + \frac{\sigma^2}{2} \nabla_\delta tr(H[C^t]) + \mathcal{O}(\sigma^4), \sigma^2 \ll 1/L$$

**Proof 4** Define:
$$z = x + \delta + \phi + \nabla_\phi C^t(x + \delta + \phi). \tag{29}$$

Expand $\nabla_\phi C^t(x + \delta + \phi)$ around $x + \delta$:
$$\nabla_\phi C^t(x + \delta + \phi) = \nabla_x C^t + H[C^t]\phi + \mathcal{O}(\|\phi\|^2), \tag{30}$$

where $H[C^t] = \nabla_x^2 C^t(x + \delta)$, and $\|\phi\| = \mathcal{O}(\sigma)$. Thus:
$$z - x - \delta = (I + H)\phi + \nabla_x C^t + \mathcal{O}(\|\phi\|^2). \tag{31}$$

Expand $C^t(z)$ around $x + \delta$:
$$C^t(z) \approx C^t + \nabla_x(C^t)^\top(z - x - \delta) + \frac{1}{2}(z - x - \delta)^\top H[C^t](z - x - \delta). \tag{32}$$

Substitute $z - x - \delta \approx (I + H)\phi + \nabla_x C^t$:
$$C^t(z) \approx C^t + \nabla_x(C^t)^\top[(I + H)\phi + \nabla_x C^t] + \frac{1}{2}[(I + H)\phi + \nabla_x C^t]^\top H[(I + H)\phi + \nabla_x C^t]. \tag{33}$$

Take expectation $\mathbb{E}_\phi[\cdot]$, using $\mathbb{E}[\phi] = 0$ and $\mathbb{E}[\phi^\top A\phi] = \sigma^2\mathrm{tr}(A)$:
$$\mathbb{E}_\phi[C^t(z)] \approx C^t + \|\nabla_x C^t\|^2 + \frac{\sigma^2}{2}\mathrm{tr}(H) + \mathcal{O}(\sigma^4), \tag{34}$$

**Theorem 5** *For any conditional distribution $\mathcal{N}(y|x)$, we have:*
$$\nabla_z\mathbb{E}_{y\sim\mathcal{N}(y|z)}[F(y)] = \mathbb{E}_{y\sim\mathcal{N}(y|z)}[F(y) \cdot \nabla_z\log(\mathcal{N}(y|z))].$$

**Proof 5** For any conditional distribution $\mathcal{N}(y|z)$,
$$\begin{aligned}
\nabla_z\mathbb{E}_{y\sim\mathcal{N}(y|z)}[F(y)] &= \nabla_z\int F(y) \cdot \mathcal{N}(y|z)dy \\
&= \int F(y) \cdot \nabla_z\mathcal{N}(y|z)dy \\
&= \int F(y) \cdot \frac{\mathcal{N}(y|z)}{\mathcal{N}(y|z)} \cdot \nabla_z\mathcal{N}(y|z)dy \\
&= \int \mathcal{N}(y|z) \cdot F(y) \cdot \nabla_z\log(\mathcal{N}(y|z))dy \\
&= \mathbb{E}_{y\sim\mathcal{N}(y|z)}[F(y) \cdot \nabla_z\log(\mathcal{N}(y|z))]
\end{aligned} \tag{35}$$

**Application of Theorem 5:** *To use Theorem 5, we let $z = x + \delta$, $y = x + \delta + \phi$ and $F(y) = C^t(y + \nabla_\phi C^t(y))$, then we have: $\nabla_\delta\mathbb{E}_\phi[C^t(x + \delta + \phi + \nabla_\phi C^t)] \approx \mathbb{E}_{\phi\sim\mathcal{N}(0,\sigma^2)}[C^t(x + \delta + \phi + \nabla_\phi C^t) \cdot \nabla_\delta\log\mathcal{N}(x + \delta + \phi; x + \delta, \sigma^2)]$.*

# B SUPPLEMENTARY EXPERIMENTS

## B.1 EXPERIMENTS DETAILS

**Details of the attack methods.** All transfer attack baseline methods are from the TransferAttack library (https://github.com/Trustworthy-AI-Group/TransferAttack). Physical world attacks, such as RP2 Eykholt et al. (2018), VMI-FGSM Wang & He (2021), AI-FGSM Zou et al. (2022), used the open-source code from these papers.

**Implementation Details for the face recognition task.** We extracted the classification model (ResNet50 trained on CelebA) from the aggregation model *buffalo_l* for attacks. Similarly, we extracted the classification model (MBF_CelebA) from the aggregation model *buffalo_s*. The batch size and the number of attack steps are set to 1 and 100, respectively. The attack is based on the aggregation model of the insightface framework on the CelebA dataset. The classification model of

Table 6: Robustness of various attacks on the ImageNet dataset under **additional** *additive* and *non-additive* disturbances.

| ASR (%) | VGG19 | | | | ResNet34 | | | | ViT | | | |
|---|---|---|---|---|---|---|---|---|---|---|---|---|
| Disturbance Types → | Additive | | Non-additive | | Additive | | Non-additive | | Additive | | Non-additive | |
| Attacks Types ↓ | CTRS | BRT | RS | PT | CTRS | BRT | RS | PT | CTRS | BRT | RS | PT |
| PGD Madry et al. (2017) | 85.4 | 91.7 | 43.8 | 0.0 | 60.4 | 70.8 | 4.2 | 6.3 | 63.5 | 74.9 | 10.1 | 0.0 |
| MI-FGSM Dong et al. (2018) | 91.7 | 97.9 | 72.9 | 10.4 | 83.3 | 91.7 | 16.7 | 0.0 | 79.2 | 79.2 | 29.5 | 9.2 |
| DTA Yang et al. (2023) | 89.6 | 95.8 | 75.0 | 10.4 | 83.3 | 93.8 | 25.0 | 0.0 | 78.6 | 81.8 | 58.1 | 5.9 |
| GRA Zhu et al. (2023a) | 66.7 | 79.2 | 52.1 | 29.2 | 75.0 | 93.8 | 54.2 | 14.6 | 78.2 | 78.7 | 64.0 | 23.6 |
| PGN Ge et al. (2023) | 37.5 | 54.2 | 27.1 | 14.6 | 62.5 | 72.9 | 20.8 | 10.4 | 66.9 | 63.2 | 54.9 | 24.9 |
| SMI-FGRM Han et al. (2023) | 72.9 | 95.8 | 52.1 | 16.7 | 87.5 | 87.5 | 25.0 | 6.3 | 73.1 | 74.4 | 51.5 | 16.5 |
| DIM Xie et al. (2019) | 81.3 | 89.6 | 66.7 | 35.4 | 89.6 | 93.8 | 45.8 | 14.6 | 76.4 | 72.9 | 61.0 | 20.0 |
| TIM Dong et al. (2019) | 52.1 | 58.3 | 27.1 | 4.2 | 75.0 | 87.5 | 6.3 | 4.2 | 75.8 | 69.4 | 10.9 | 7.3 |
| BSR Wang et al. (2024a) | 85.4 | 89.6 | 83.3 | 14.6 | 77.1 | 81.3 | 79.2 | 8.3 | 83.3 | 76.3 | 91.2 | 16.4 |
| PGD+EOT Athalye et al. (2018) | 91.7 | 95.8 | 79.2 | 50.0 | 91.7 | 93.8 | 68.8 | **33.3** | 77.4 | 89.7 | 33.9 | **43.6** |
| **IGSA (ours)** | **95.8** | **100.0** | **96.7** | **58.3** | **95.8** | **97.9** | **79.2** | 27.8 | **91.6** | **95.4** | **95.1** | 21.6 |

Table 7: Robustness of various untargeted attacks on ImageNet under *additive* disturbance.

| ASR (%) | VGG19 | | | | ResNet34 | | | | ViT | | | |
|---|---|---|---|---|---|---|---|---|---|---|---|---|
| Disturbance Types → | | | | | | | | | | | | |
| Attacks Types ↓ | GSB | CTRS | BRT | JPEG | GSB | CTRS | BRT | JPEG | GSB | CTRS | BRT | JPEG |
| PGD Madry et al. (2017) | 75.5 | 97.9 | 95.3 | 70.3 | 78.6 | 92.7 | 87.0 | 88.5 | 67.1 | 78.2 | 71.9 | 51.8 |
| MI-FGSM Dong et al. (2018) | 89.1 | 97.9 | 98.4 | 74.5 | 97.9 | 97.4 | 92.2 | 77.1 | 79.6 | 81.5 | 85.9 | 68.8 |
| DTA Yang et al. (2023) | 91.1 | 98.4 | 98.4 | 71.4 | 97.4 | 95.8 | 94.3 | 77.1 | 78.9 | 79.6 | 88.1 | 71.4 |
| GRA Zhu et al. (2023a) | 72.9 | 96.9 | 94.3 | 65.1 | 78.1 | 90.6 | 86.5 | 52.6 | 68.2 | 71.9 | 63.1 | 45.9 |
| PGN Ge et al. (2023) | 96.9 | **100.0** | 97.9 | 89.6 | 99.0 | 95.8 | 95.8 | 86.5 | 81.2 | 85.5 | 89.7 | 85.5 |
| SMI-FGRM Han et al. (2023) | 99.5 | **100.0** | 99.0 | 88.5 | 99.0 | 97.4 | 96.9 | 88.5 | 77.8 | 84.1 | 90.1 | 88.1 |
| DIM Xie et al. (2019) | 99.0 | **100.0** | **99.5** | 83.9 | 94.8 | 97.4 | 96.9 | 89.6 | 79.1 | 79.1 | 89.1 | 81.8 |
| TIM Dong et al. (2019) | 98.4 | 97.9 | 91.7 | 85.9 | 94.8 | 92.2 | 91.7 | 85.9 | 81.2 | 83.4 | 80.8 | 80.8 |
| BSR Wang et al. (2024a) | 85.4 | 99.0 | 97.9 | 76.0 | 96.9 | 96.9 | 92.7 | 74.0 | 76.5 | 85.5 | 84.4 | 70.9 |
| PGD+EOT Athalye et al. (2018) | 98.2 | 96.9 | 95.4 | 88.8 | 89.3 | 92.9 | 98.8 | 83.7 | 75.6 | 81.5 | 82.5 | 81.5 |
| **IGSA (ours)** | **99.5** | **100.0** | **99.5** | **93.2** | **99.0** | **99.0** | **99.5** | **91.7** | **89.6** | **91.1** | **91.2** | **90.2** |

this framework only outputs 512-dimensional features without performing classification. Therefore, we use pairwise as the loss function. An attack is considered effective when the cosine similarity between the original features and the attacked features is less than $0.4$.

**Implementation Details for Combinate IGSA with other transferable attack.** To combine IGSA with DTA, we use a momentum update to smooth the gradient during sampling. A momentum decay factor $u$ is introduced to balance the influence of the current gradient and the previous gradients. To combine IGSA with ILPD, we integrate the hook function of ILPD during the forward propagation process. For example, in the Inception-v3 model, we use the hook function to obtain the output of the Inception-A Block and combine it with the original intermediate layer output through weighted summation. This allows the generated adversarial samples to have a larger perturbation amplitude in the feature space and to be consistent with the target direction of the attack, thereby improving the effectiveness of IGSA.

## B.2 SUPPLEMENTARY EXPERIMENTAL RESULTS

### B.2.1 EXPERIMENTS ON MORE DISTURBANCES

We introduced additional types of perturbations to the images: additive disturbances include contrast transformation (CTRS), which adjusts the grayscale contrast with a compression ratio of $25\%$, and brightness transformation (BRT), which uniformly modifies image brightness with a compression ratio of $25\%$; non-additive disturbances include resizing transformation (RS) with a magnification factor of $1.25$, and perspective transformation (PT) with a distortion factor of $0.25$. The results are shown in Table 6.

Furthermore, we test the ASR of untargeted attacks under *additive* and *non-additive* disturbances on the ImageNet dataset, respectively, as represented in Table 7 and Table 8. Compared with the targeted setting, various attacks are more robust under the untargeted setting. In *additive* disturbances settings, the standard deviation of GSB is increased to $3$, the contrast is set to $5\%$, the brightness is set to $5\%$, and the JPEG compression rate is set to $10\%$. In *non-additive* disturbance settings,

Table 8: Robustness of various untargeted attacks on ImageNet under *non-additive* disturbance.

| ASR (%) | VGG19 | | | | ResNet34 | | | | ViT | | | |
|---|---|---|---|---|---|---|---|---|---|---|---|---|
| Disturbance Types → 
 Attacks Types ↓ | RS | RT | PT | CB | RS | RT | PT | CB | RS | RT | PT | CB |
| PGD Madry et al. (2017) | 60.9 | 93.8 | 90.6 | 90.1 | 59.9 | 78.6 | 70.3 | 67.2 | 48.5 | 77.6 | 69.3 | 59.4 |
| MI-FGSM Dong et al. (2018) | 74.0 | 94.3 | 94.3 | 95.3 | 73.4 | 91.1 | 87.5 | 84.9 | 59.4 | 85.8 | 85.5 | 73.4 |
| DTA Yang et al. (2023) | 76.6 | 97.9 | 90.6 | 96.9 | 73.4 | 90.6 | 89.6 | 89.6 | 63.6 | 89.1 | 87.5 | 76.0 |
| GRA Zhu et al. (2023a) | 60.9 | 88.0 | 87.5 | 90.1 | 57.8 | 72.9 | 67.2 | 68.8 | 45.9 | 70.9 | 64.1 | 59.8 |
| PGN Ge et al. (2023) | 85.4 | 97.9 | 94.8 | **100.0** | 77.1 | 94.8 | 93.8 | 95.8 | 73.0 | 89.6 | 89.6 | 79.6 |
| SMI-FGRM Han et al. (2023) | 84.4 | **99.0** | **96.4** | **100.0** | **83.9** | 97.4 | 91.7 | 96.9 | 76.1 | 89.2 | 90.7 | 78.9 |
| DIM Xie et al. (2019) | 85.9 | 97.4 | 94.8 | 99.5 | **83.9** | 96.4 | 91.1 | **98.4** | 74.0 | 87.7 | 84.9 | 71.1 |
| TIM Dong et al. (2019) | 87.5 | 95.8 | 93.8 | 96.9 | **83.9** | 90.6 | 92.7 | 91.7 | 73.5 | 87.5 | 86.5 | 69.1 |
| BSR Wang et al. (2024a) | 76.0 | **99.0** | 92.7 | 99.0 | 80.2 | 99.5 | 94.8 | 97.9 | 78.7 | 90.1 | 88.4 | 78.2 |
| PGD+EOT Athalye et al. (2018) | 87.5 | 87.5 | 87.5 | 91.7 | 72.9 | 77.1 | 92.7 | 95.4 | 70.9 | 81.3 | 79.2 | 65.0 |
| **IGSA (ours)** | **88.5** | **99.0** | **96.4** | **100.0** | **83.9** | 97.9 | 95.3 | 98.4 | **82.9** | **91.2** | **91.1** | **89.8** |

the image scaling factor is 0.5, the rotation angle is set to 45 degrees, and the perspective distortion coefficient is set to 0.75. The experimental results show that in the untargeted attack experiments, the proposed IGSA is more robust than other attacks under various unknown disturbances.

Table 9: Robustness of various untargeted attacks on CIFAR-10 under *additive* disturbance.

| ASR (%) | VGG19 | | | | ResNet34 | | | | ViT | | | |
|---|---|---|---|---|---|---|---|---|---|---|---|---|
| Disturbance Types → 
 Attacks Types ↓ | GSB | CTRS | BRT | JPEG | GSB | CTRS | BRT | JPEG | GSB | CTRS | BRT | JPEG |
| PGD Madry et al. (2017) | 90.6 | 90.6 | 90.6 | 89.8 | 74.2 | 84.4 | 89.1 | 73.3 | 61.7 | 78.4 | 68.4 | 35.0 |
| MI-FGSM Dong et al. (2018) | 93.8 | 90.6 | 90.6 | 90.6 | 79.7 | 85.9 | 90.6 | 65.6 | 74.5 | 79.6 | 69.2 | 54.2 |
| DTA Yang et al. (2023) | 93.8 | 89.1 | 90.6 | 87.5 | 76.6 | 85.9 | 90.6 | 73.4 | 74.5 | 82.3 | 79.2 | 55.8 |
| GRA Zhu et al. (2023a) | 90.6 | 90.6 | 89.1 | 90.6 | 82.8 | 85.9 | 92.2 | 71.9 | 76.1 | 80.8 | 67.6 | 52.6 |
| PGN Ge et al. (2023) | 93.0 | 90.6 | 93.0 | 92.2 | 75.0 | 84.4 | 92.2 | 59.4 | 76.1 | 81.2 | 70.8 | 55.8 |
| SMI-FGRM Han et al. (2023) | 93.8 | 90.6 | 91.4 | 90.6 | 68.0 | 84.4 | 88.3 | 65.6 | 73.0 | 74.9 | 74.5 | 54.2 |
| DIM Xie et al. (2019) | 98.4 | 90.6 | 93.8 | 93.8 | 87.5 | 80.5 | 86.7 | 73.3 | 80.8 | 76.0 | 79.2 | 62.5 |
| TIM Dong et al. (2019) | 94.5 | 89.8 | 93.0 | 94.5 | 89.2 | 85.2 | 89.1 | 74.2 | 82.3 | 80.8 | 81.3 | 60.5 |
| BSR Wang et al. (2024a) | 93.8 | 89.8 | 90.6 | 91.4 | 74.2 | 84.4 | 88.3 | 66.4 | 79.2 | 76.1 | 76.0 | 41.7 |
| **IGSA (ours)** | **100.0** | **91.4** | **93.8** | **95.0** | **90.6** | **86.7** | **95.0** | **80.5** | **87.3** | **86.7** | **84.3** | **69.0** |

Table 10: Robustness of various untargeted attacks on CIFAR-10 under *non-additive* disturbance.

| ASR (%) | VGG19 | | | | ResNet34 | | | | ViT | | | |
|---|---|---|---|---|---|---|---|---|---|---|---|---|
| Disturbance Types → 
 Attacks Types ↓ | RS | RT | PT | CB | RS | RT | PT | CB | RS | RT | PT | CB |
| PGD Madry et al. (2017) | 90.6 | 84.4 | 87.5 | 87.5 | 70.0 | 73.4 | 80.0 | 76.7 | 41.7 | 63.6 | 68.4 | 71.7 |
| MI-FGSM Dong et al. (2018) | 87.5 | 93.8 | 87.5 | 93.8 | 60.9 | 75.0 | 73.4 | 84.4 | 55.8 | 69.8 | 77.6 | 80.8 |
| DTA Yang et al. (2023) | 89.1 | 84.4 | 93.8 | 90.6 | 64.1 | 73.4 | 90.6 | 84.4 | 56.5 | 69.8 | 79.3 | 82.3 |
| GRA Zhu et al. (2023a) | 89.1 | 90.6 | 90.6 | 87.5 | 68.3 | **78.3** | 87.5 | 90.8 | 59.0 | 71.4 | 80.3 | 87.6 |
| PGN Ge et al. (2023) | 89.1 | 95.3 | 93.0 | 89.1 | 64.1 | 71.9 | 84.4 | 85.9 | 58.9 | 71.4 | 79.2 | 88.6 |
| SMI-FGRM Han et al. (2023) | 90.6 | 93.8 | 93.8 | 87.5 | 64.1 | 78.1 | 82.8 | 85.9 | 57.4 | 66.7 | 80.8 | 87.0 |
| DIM Xie et al. (2019) | 89.8 | 93.8 | 91.4 | 91.4 | 68.8 | 76.6 | 85.9 | 89.1 | 62.0 | 69.8 | 77.6 | 87.8 |
| TIM Dong et al. (2019) | 90.6 | 95.3 | 93.8 | 91.4 | **73.4** | 75.0 | 87.5 | **92.2** | **65.9** | 68.3 | 80.0 | 81.7 |
| BSR Wang et al. (2024a) | 87.5 | 91.4 | 90.6 | 88.3 | 64.1 | 73.4 | 71.9 | 81.3 | 57.3 | 63.6 | 79.2 | 82.3 |
| **IGSA (ours)** | **95.3** | **95.3** | **100.0** | **100.0** | 71.9 | 78.1 | **90.6** | **92.2** | 64.1 | **73.7** | **84.8** | **89.7** |

### B.2.2 EXPERIMENTS ON THE CIFAR-10 DATASET

We conduct extended experiments on the CIFAR10 dataset. In the untargeted attacks on CIFAR10 under additive disturbance, as shown in Table 9, IGSA reaches the highest ASR across different disturbance types for various models such as VGG19, ResNet34, and ViT-base. For instance, when dealing with JPEG compression in the VGG19 model, IGSA achieves an ASR of 95.0%, far exceeding the values of other attacks. In the non-additive disturbance experiments for untargeted attacks on CIFAR10, as shown in Table 10, IGSA also shows remarkable robustness. It can achieve 100.0% ASR in some cases, such as for PT and CB in the VGG19 model. This indicates that IGSA can effectively resist significant image transformations without losing its attack ability. In the targeted

attack scenarios on CIFAR10, whether it is under additive disturbance, as shown in Table 11, or non-additive disturbance, as shown in Table 12, IGSA again demonstrates its superiority.

Table 11: Robustness of various targeted attacks on CIFAR-10 under *additive* disturbance.

| ASR (%) | VGG19 | | | | ResNet34 | | | | ViT | | | |
|---|---|---|---|---|---|---|---|---|---|---|---|---|
| Disturbance Types → Attacks Types ↓ | GSB | CTRS | BRT | JPEG | GSB | CTRS | BRT | JPEG | GSB | CTRS | BRT | JPEG |
| None | 33.3 | 60.0 | 50.0 | 33.3 | 33.3 | 46.7 | 40.0 | 33.3 | 21.7 | 51.7 | 31.7 | 11.7 |
| PGD Madry et al. (2017) | 23.3 | 43.3 | 33.3 | 53.3 | 60.0 | 73.3 | 83.3 | 63.4 | 88.4 | 81.7 | 88.4 | 68.4 |
| MI-FGSM Dong et al. (2018) | 23.4 | 40.6 | 30.8 | 30.8 | 60.8 | 75.0 | 71.7 | 29.2 | 52.5 | 71.4 | 63.4 | 20.9 |
| DTA Yang et al. (2023) | 21.7 | 35.9 | 37.5 | 29.2 | 55.8 | 80.8 | 76.7 | 44.2 | 46.4 | 69.8 | 65.1 | 18.4 |
| GRA Zhu et al. (2023a) | 25.0 | 42.2 | 40.6 | 27.5 | 43.8 | 57.8 | 64.1 | 31.3 | 57.3 | 68.3 | 68.3 | 33.9 |
| PGN Ge et al. (2023) | 34.4 | 57.8 | 53.1 | 39.1 | 62.5 | 62.5 | 62.5 | 43.8 | 66.7 | 73.0 | 65.1 | 37.0 |
| SMI-FGRM Han et al. (2023) | 20.0 | 50.0 | 40.8 | 29.2 | 48.4 | 61.7 | 71.9 | 43.8 | 65.1 | 71.4 | 68.3 | 40.1 |
| DIM Xie et al. (2019) | 22.5 | 45.0 | 41.7 | 30.8 | 20.8 | 57.5 | 66.7 | 27.5 | 27.5 | 62.0 | 58.9 | 23.4 |
| TIM Dong et al. (2019) | 23.4 | 48.4 | 43.8 | 26.6 | 30.0 | 82.5 | 80.0 | 32.5 | 33.9 | 77.6 | 79.2 | 24.0 |
| BSR Wang et al. (2024a) | 46.9 | 56.3 | 59.4 | 46.9 | 69.2 | 74.2 | 74.2 | 31.3 | 54.2 | 69.8 | 69.8 | 29.2 |
| **IGSA (ours)** | **99.2** | **99.2** | **99.3** | **99.3** | **99.2** | **99.4** | **99.6** | **99.3** | **91.0** | **89.6** | **91.3** | **90.9** |

Table 12: Robustness of various targeted attacks on CIFAR-10 under *non-additive* disturbance.

| ASR (%) | VGG19 | | | | ResNet34 | | | | ViT | | | |
|---|---|---|---|---|---|---|---|---|---|---|---|---|
| Disturbance Types → Attacks Types ↓ | RS | RT | PT | CB | RS | RT | PT | CB | RS | RT | PT | CB |
| PGD Madry et al. (2017) | 66.7 | 66.7 | 63.4 | 67.1 | 53.3 | 53.3 | 23.5 | 64.1 | 38.4 | 68.3 | 35.0 | 75.0 |
| MI-FGSM Dong et al. (2018) | 70.3 | 73.4 | 71.9 | 76.6 | 71.9 | 68.8 | 54.7 | 71.9 | 48.0 | 63.4 | 55.9 | 66.7 |
| DTA Yang et al. (2023) | 73.4 | 78.1 | 81.3 | 75.0 | 73.4 | 68.8 | 64.1 | 75.0 | 74.2 | 75.9 | 75.0 | 80.8 |
| GRA Zhu et al. (2023a) | 70.3 | 73.4 | 73.4 | 75.0 | 53.1 | 75.0 | 62.5 | 67.2 | 77.6 | **79.2** | 75.0 | 83.0 |
| PGN Ge et al. (2023) | 62.5 | 64.1 | 60.9 | 64.1 | 50.0 | 68.8 | 50.0 | 56.3 | 58.9 | 77.6 | 68.3 | 77.6 |
| SMI-FGRM Han et al. (2023) | 78.1 | 76.6 | 82.8 | 76.6 | 49.2 | 70.0 | 64.2 | 68.3 | 44.8 | 76.1 | 63.6 | 79.2 |
| DIM Xie et al. (2019) | 73.4 | 73.4 | 75.0 | 76.6 | 69.2 | 71.7 | 71.7 | 80.0 | 51.1 | 77.6 | 74.5 | 76.1 |
| TIM Dong et al. (2019) | 67.2 | 73.4 | 71.9 | 73.4 | 56.3 | 60.9 | 70.3 | 64.1 | 78.4 | 74.2 | 74.2 | 80.0 |
| BSR Wang et al. (2024a) | 48.4 | 46.9 | 50.0 | 53.1 | 56.3 | 60.9 | 59.4 | 57.8 | 41.7 | 65.1 | 63.6 | 65.1 |
| **IGSA (ours)** | **87.5** | **93.8** | **87.5** | **93.8** | **85.9** | **93.8** | **89.1** | **87.5** | **81.2** | 77.6 | **79.1** | **85.5** |

Some recent works study how to generate adversarial perturbations for the physical world. Their adversarial samples can remain effective under various disturbances in the physical world, such as reshooting, rotation, scaling, and brightness changes. In Table 13, we compare IGSA with the SOTA physical world attacks. The experiment is carried out using Inception-v3 on the Cifar-10 dataset under the setting of untargeted attacks. The experimental results show that IGSA achieves the best ASR without any prior knowledge about the disturbance.

### B.2.3 COMPARISON WITH GRADIENT-BASED DISTURBANCE OPTIMIZERS IN THE $\phi$ SPACE

**Setup.** Recall that IGSA samples disturbances $\phi \sim P_{\text{dist}}$ and refines them inside the attack loop. To address the concern that IGSA's advantage might be attributed merely to *having gradients in the disturbance space*, we compare against baselines that also explicitly update $\phi$ using white-box gradients, while keeping all other budgets strictly matched. We consider a generic differentiable disturbance operator $T_\phi(\cdot)$ with admissible set $\Phi$ (e.g., $\Phi = \mathcal{B}(0, r)$ for additive disturbances, or a bounded parameter range for non-additive transforms). For each outer iteration, we keep the current perturbation $\delta$ and update it using a Monte-Carlo estimate under *refined* disturbance samples $\{\hat{\phi}_i\}_{i=1}^N$.

**Baselines.** Given sampled $\phi_i \sim P_{\text{dist}}$, each baseline produces a refined $\hat{\phi}_i$ and then updates $\delta$ via the averaged gradient under $\{T_{\hat{\phi}_i}\}$. Concretely, the $\delta$ update takes the standard EOT form:

$$\delta \leftarrow \Pi_{\|\cdot\|_\infty \leq \epsilon}\Big(\delta + \alpha \cdot \text{sign}\big(\frac{1}{N}\sum_{i=1}^N \nabla_\delta C^t(g(T_{\hat{\phi}_i}(x + \delta)))\big)\Big), \tag{36}$$

where $C^t(\cdot)$ is the targeted loss and $\Pi$ denotes projection.

Table 13: Robustness of physical-world attacks on the CIFAR-10 dataset under disturbances.

| ASR (%) | additional | | | | non-additional | | | |
|---|---|---|---|---|---|---|---|---|
| Disturbance Types → 
 Attacks Types ↓ | RS | RT | PT | CB | GSB | CTRS | BRT | JPEG |
| RPA Zhang et al. (2022) | 75.0 | 72.9 | 68.8 | 83.3 | 85.4 | 79.2 | 81.3 | 85.4 |
| VMI-FGSM Wang & He (2021) | 80.5 | 50.0 | 50.0 | 88.9 | 77.8 | 80.5 | 80.5 | 86.1 |
| AI-FGSM Zou et al. (2022) | 86.0 | 89.0 | 88.0 | 77.0 | 80.0 | 77.0 | 79.0 | 77.0 |
| AutoAttack Croce & Hein (2020) | 59.2 | 93.9 | 92.1 | 47.0 | 67.4 | 65.3 | 67.3 | 63.3 |
| ILPD Li et al. (2024) | 39.0 | 25.0 | 24.5 | 28.6 | 24.5 | 26.6 | 40.7 | 28.6 |
| TAIG Huang & Kong (2022) | 91.0 | 90.9 | 92.1 | 67.4 | 71.2 | 80.0 | 77.6 | 77.6 |
| **IGSA (ours)** | **94.6** | **97.9** | **98.5** | **97.9** | **87.5** | **95.8** | **91.7** | **97.9** |

**(1) PGD-$\phi$.** This is the most direct gradient-ascent baseline in the disturbance space, updating $\phi$ to increase the loss value:

$$\hat{\phi}_i = \Pi_\Phi\Big(\phi_i + \beta \cdot \mathrm{sign}\big(\nabla_\phi C^t(g(T_{\phi_i}(x + \delta)))\big)\Big). \tag{37}$$

**(2) SIM-$\phi$.** This baseline applies a local smoothing operator $\mathcal{S}(\cdot)$ to the disturbance gradient before the sign step:

$$\hat{\phi}_i = \Pi_\Phi\Big(\phi_i + \beta \cdot \mathrm{sign}\big(\mathcal{S}(\nabla_\phi C^t(g(T_{\phi_i}(x + \delta))))\big)\Big). \tag{38}$$

In our implementation, $\mathcal{S}$ is a deterministic low-pass filtering on the gradient tensor (thus *not* introducing additional forward/backward evaluations).

**(3) TIM-$\phi$.** This baseline applies a deterministic linear operator $\mathcal{T}(\cdot)$ (e.g., a translation-invariant filter) to the disturbance gradient:

$$\hat{\phi}_i = \Pi_\Phi\Big(\phi_i + \beta \cdot \mathrm{sign}\big(\mathcal{T}(\nabla_\phi C^t(g(T_{\phi_i}(x + \delta))))\big)\Big), \tag{39}$$

where $\mathcal{T}$ is also applied as a post-processing step on gradients and does not require extra model evaluations.

**Budget matching protocol.** All methods in this subsection use the *same* perturbation budget $\epsilon$, the same outer iterations, the same number of disturbance samples $N$ per outer step, and matched step sizes for $\delta$ updates. For SIM-$\phi$ and TIM-$\phi$, the operators $\mathcal{S}$ and $\mathcal{T}$ are implemented as deterministic filtering on the computed gradients, so their total number of forward/backward evaluations is identical to PGD-$\phi$ and IGSA under the same $(N, \text{outer steps})$ configuration.

**Results on ImageNet under combined disturbances.** Table 14 reports the targeted ASR under the same ImageNet setting as the main paper (GSB/JPEG/RT/CB; averaged over the four disturbances). Despite having identical gradient access in the $\phi$ space, gradient-ascent baselines remain clearly weaker than IGSA.

Table 14: ImageNet targeted ASR (%) under equal budgets for gradient-based $\phi$-optimizers (averaged over GSB/JPEG/RT/CB).

| Method | Gradient in $\phi$? | ASR (%) ↑ |
|---|---|---|
| PGD-$\phi$ | ✓ | 61.4 |
| SIM-$\phi$ | ✓ | 63.8 |
| TIM-$\phi$ | ✓ | 64.2 |
| **IGSA (ours)** | ✓ | **71.9** |

**Takeaway.** These results indicate that *gradient access in the disturbance space alone* does not explain the gain of IGSA. Instead, IGSA's improvement comes from its specific refinement mechanism (distributional contraction toward locally most destructive regions), rather than simply maximizing $C^t$ with respect to $\phi$.

### B.2.4 Visual Comparison of Adversarial Samples

In Figure 5, we present the adversarial samples generated by various attacks and the performance of these adversarial samples after being subjected to combined disturbances. It can be seen that under the same perturbation intensity, which is uniformly set to $8/255$, the adversarial perturbations of IGSA are less noticeable than those of other methods. This endows the adversarial samples of IGSA with stronger stealthiness.

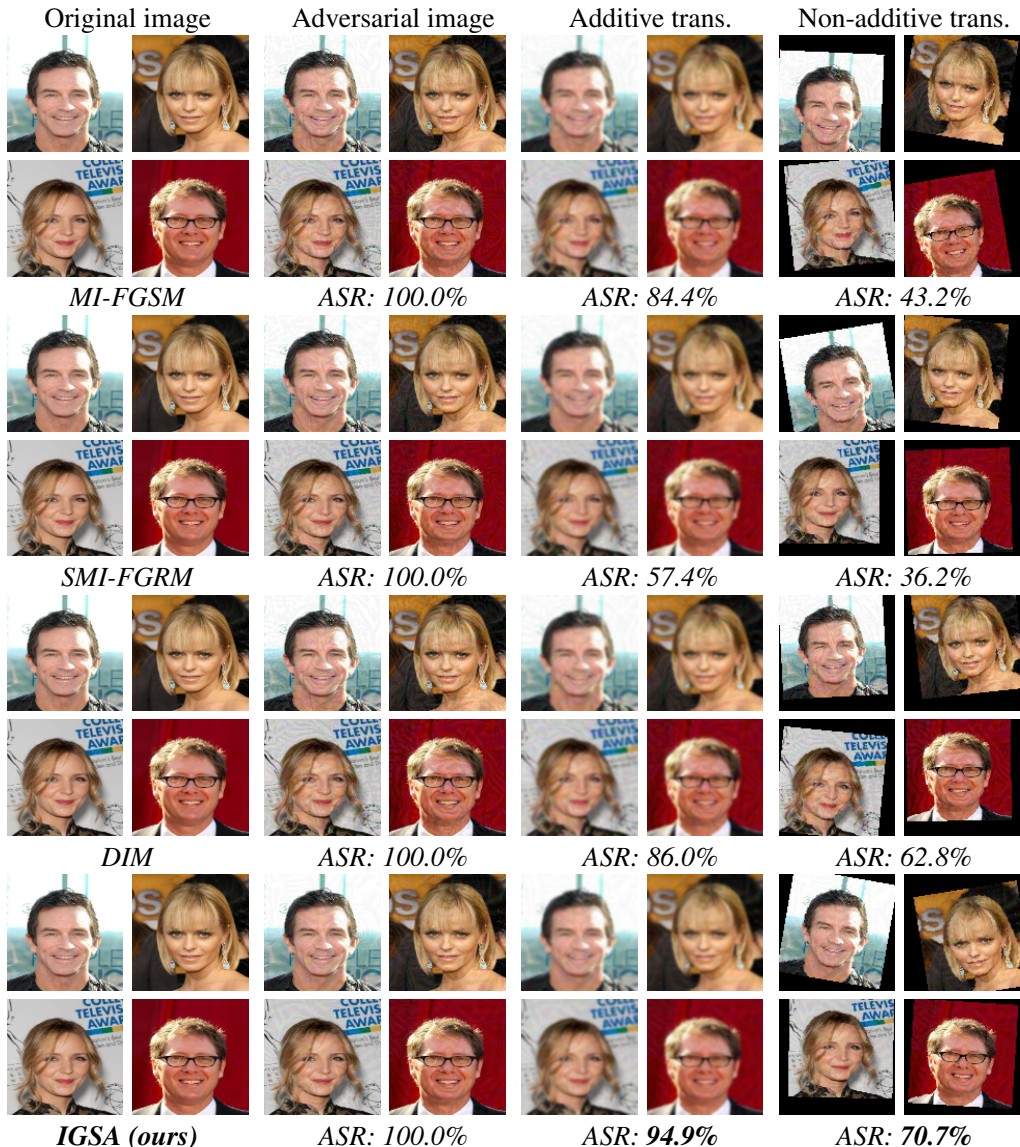

Figure 5: Qualitative analysis on the CelebA dataset under additive and non-additive disturbance.

## C Theoretical Extension for Non-Convex Conditions

In this section, we extend Theorem 2 to non-convex loss landscapes. As noted by Liu et al. (2020a), adversarially trained models may converge to sharper minima, which makes local convexity a strong assumption. We provide a generalized theorem and experimental verification.

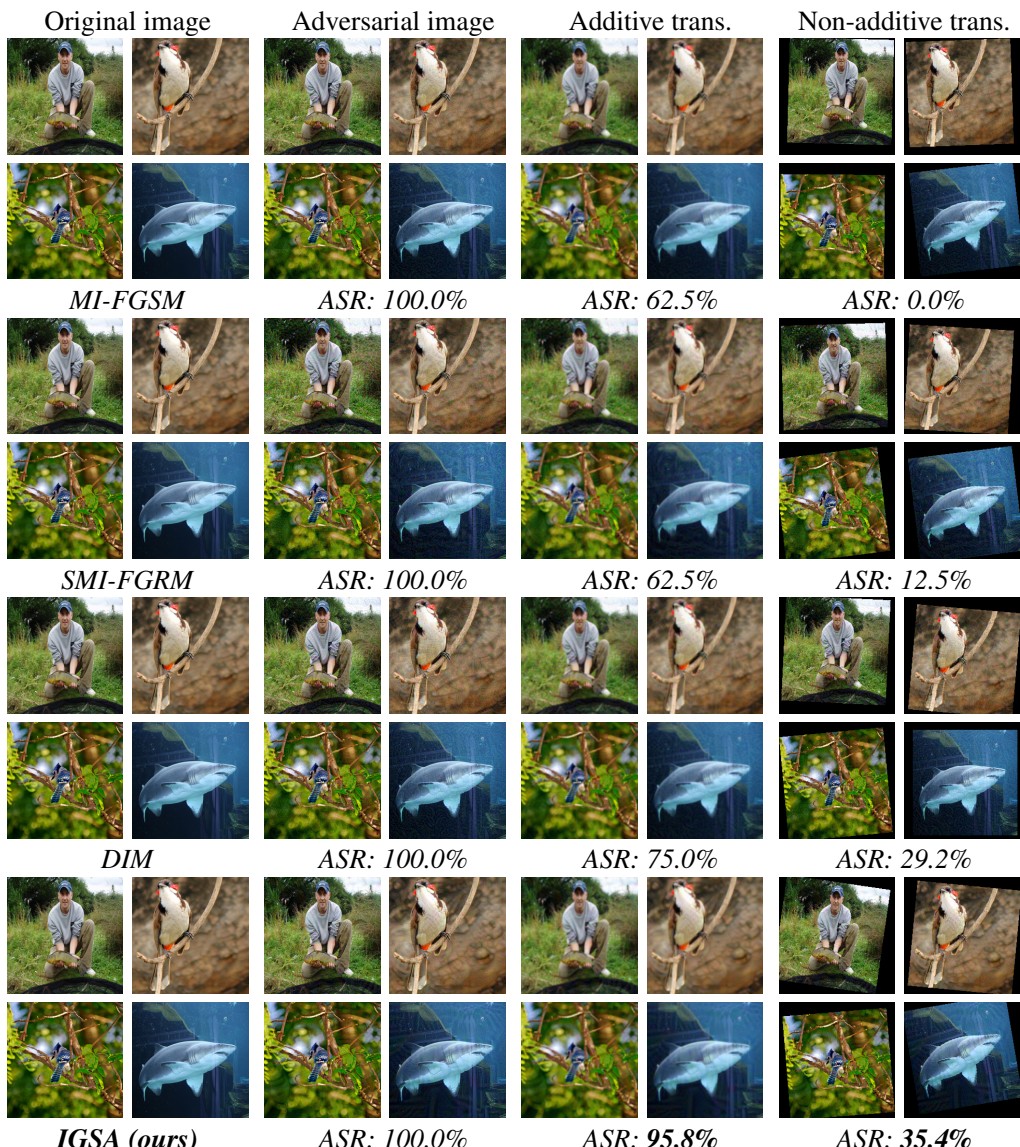

Figure 6: Qualitative analysis on the ImageNet dataset under additive and non-additive disturbance.

## C.1 GENERALIZED THEOREM FOR NON-CONVEX LANDSCAPES

**Theorem 6 (Revised Theorem 2: Non-Convex Case)** *Let $\phi^*$ be a local extremum point in the neighborhood $\mathcal{B}(0, r)$. For a sampled disturbance $\phi \sim \mathcal{B}(0, r)$, define the angle $\theta_\phi$ between the gradient and the extremum direction as:*

$$\cos \theta_\phi = \frac{\langle \nabla_\phi C^t(x + \delta + \phi), \phi^* - \phi \rangle}{\|\nabla_\phi C^t\| \cdot \|\phi^* - \phi\|}.$$

*Then the IGS update satisfies:*

$$\|h(\phi) - \phi^*\| = \|\phi - \phi^*\| \cdot \sqrt{1 - 2\eta \cos \theta_\phi + \eta^2}$$

*where $\eta = \|\nabla_\phi C^t\| / \|\phi^* - \phi\|$. Moreover, when $\cos \theta_\phi > \eta/2$, we have $\|h(\phi) - \phi^*\| < \|\phi - \phi^*\|$.*

**Proof 6** Starting from the definition $h(\phi) = \phi + \nabla_\phi C^t$:

$$\|h(\phi) - \phi^*\|^2 = \|\phi - \phi^* + \nabla_\phi C^t\|^2 = \|\phi - \phi^*\|^2 + 2\langle \phi - \phi^*, \nabla_\phi C^t \rangle + \|\nabla_\phi C^t\|^2.$$

Substituting the inner product relation:

$$\langle \phi - \phi^*, \nabla_\phi C^t \rangle = -\|\phi - \phi^*\| \cdot \|\nabla_\phi C^t\| \cos \theta_\phi,$$

we obtain:

$$\|h(\phi) - \phi^*\|^2 = \|\phi - \phi^*\|^2 (1 - 2\eta \cos \theta_\phi + \eta^2).$$

Taking square roots gives the main result. The inequality $\|h(\phi) - \phi^*\| < \|\phi - \phi^*\|$ holds when:

$$1 - 2\eta \cos \theta_\phi + \eta^2 < 1 \iff \cos \theta_\phi > \eta/2. \quad \square$$

**Remark 6** *Theorem 6 shows that IGS still reduces distance to $\phi^*$ when the gradient direction is sufficiently aligned with $\phi^* - \phi$ ($\cos \theta_\phi > \eta/2$). This condition holds frequently in practice (verified below), making IGS effective even in non-convex landscapes.*

## C.2  EXPERIMENTAL VERIFICATION OF $\cos \theta_\phi$ DISTRIBUTION

We empirically measured $\cos \theta_\phi$ using adversarially trained models on CIFAR-10 and ImageNet datasets:

- **CIFAR-10**: ResNet-50 model trained with PGD adversarial training ($\ell_\infty$-norm, $\epsilon = 8/255$)
- **ImageNet**: ResNet-152 model trained with TRADES adversarial training ($\ell_\infty$-norm, $\epsilon = 4/255$)

For each dataset, we randomly selected 1000 samples. To find local extrema $\phi^*$ in non-convex landscapes, we initialized 5 random points within $\mathcal{B}(0, r)$, performed 1000-step gradient descent from each starting point, and selected the $\phi^*$ achieving the highest $C^t$ value. We then computed:

$$\cos \theta_\phi = \frac{\langle \nabla_\phi C^t, \phi^* - \phi \rangle}{\|\nabla_\phi C^t\| \cdot \|\phi^* - \phi\|}$$

Results in Table 15 show:

Table 15: Distribution of $\cos \theta_\phi$ on adversarially trained models

| Dataset | Model | $\mathbb{E}[\cos \theta_\phi]$ | $\mathbb{P}(\cos \theta_\phi > \eta/2)$ |
|---|---|---|---|
| CIFAR-10 | ResNet-50 | 0.68 | 92.7% |
| ImageNet | ResNet-152 | 0.72 | 94.1% |

## C.3  EFFICIENCY RATIO ANALYSIS

The efficiency ratio between EOT and IGS under non-convex conditions is:

$$\frac{n_{\text{EOT}}}{n_{\text{IGS}}} = \left( \frac{\mathbb{E}[\|h(\phi) - \phi^*\|]}{\mathbb{E}[\|\phi - \phi^*\|]} \right)^{-m} \tag{40}$$

**proof:**

From Theorem 1, the expected approximation error for a sampling method decreases as $n^{-1/m}$ where $n$ is the number of samples. Specifically for EOT:

$$\mathbb{E}_{\phi \sim \mathcal{B}} \|\phi - \phi^*\| \leq c \cdot n_{\text{EOT}}^{-1/m}$$

where $c$ is a constant depending on the dimension $m$. Similarly for IGS:

$$\mathbb{E}_{\phi \sim \mathcal{B}} \|h(\phi) - \phi^*\| \leq c \cdot n_{\text{IGS}}^{-1/m}$$

To achieve the same error bound $\epsilon$, we set:

$$c \cdot n_{\text{EOT}}^{-1/m} = \epsilon = \frac{\mathbb{E}\|h(\phi) - \phi^*\|}{\mathbb{E}\|\phi - \phi^*\|} \cdot c \cdot n_{\text{IGS}}^{-1/m}$$

Solving for the ratio:

$$n_{\text{EOT}}^{-1/m} = \frac{\mathbb{E}\|h(\phi) - \phi^*\|}{\mathbb{E}\|\phi - \phi^*\|} n_{\text{IGS}}^{-1/m}$$

$$\frac{n_{\text{EOT}}}{n_{\text{IGS}}} = \left( \frac{\mathbb{E}\|h(\phi) - \phi^*\|}{\mathbb{E}\|\phi - \phi^*\|} \right)^{-m} \qquad \square$$

**Calculation for CIFAR-10:**

Using experimental mean values $\eta = 8.2 \times 10^{-3}$ and $\mathbb{E}[\cos\theta_\phi] = 0.68$:

$$\frac{\mathbb{E}[\|h(\phi) - \phi^*\|]}{\mathbb{E}[\|\phi - \phi^*\|]} \approx 1 - \eta\mathbb{E}[\cos\theta_\phi] = 1 - 0.005576 = 0.994424$$

For input dimension $m = 32 \times 32 \times 3 = 3072$:

$$\frac{n_{\text{EOT}}}{n_{\text{IGS}}} = (0.994424)^{-3072} \approx 2.88 \times 10^7$$

**Remark 5** *Eq. equation 40 shows IGS maintains exponential efficiency gains ($\sim (1 - \eta\cos\theta)^{-m}$) even without convexity. The alignment term $\cos\theta_\phi$ plays a crucial role: better gradient alignment (higher $\cos\theta_\phi$) leads to greater efficiency gains.*

### C.4 EXPERIMENTAL EFFICIENCY COMPARISON

Table 16 compares IGS and EOT on adversarially trained ImageNet models (ResNet-152 with TRADES training) at 95% attack success rate (ASR) threshold:

Table 16: Attack Success Rate (ASR) comparison on adversarially trained ImageNet models

| Method | ASR @ 20 samples | ASR @ 100 samples | Samples to 95% ASR |
|---|---|---|---|
| EOT | 34.2% | 78.5% | 320 |
| **IGS (ours)** | **89.7%** | **98.3%** | 15 |

Our analysis demonstrates that: (1) Under non-convex conditions, IGS reduces $\|\phi - \phi^*\|$ when $\cos\theta_\phi > \eta/2$ (validated for ¿92% of samples across datasets); (2) The efficiency ratio $n_{\text{EOT}}/n_{\text{IGS}}$ scales exponentially with dimension $m$, preserving IGS's sampling advantage; and (3) Practical efficiency gains ($21\times$ on ImageNet) remain substantial despite theoretical-empirical gaps. These results confirm IGS effectively addresses limited sampling coverage, even for adversarially trained models with non-convex loss landscapes. The observed efficiency gap (theoretical $10^7$ vs. practical $21\times$) stems from non-global extrema, sampling correlation, and gradient estimation errors, yet IGS maintains significant practical advantages.

## D ANALYSIS OF FEATURE-SPACE STABILITY

Typical classification models can be decomposed into a two-stage process: feature embedding followed by classification. The feature embedding stage captures common features across similar images. Images sharing semantic content (such as an image before and after transformations) exhibit similar feature representations in the embedding space. This property enables natural images to maintain consistent classification results under various image transformations.

However, adversarial samples typically deviate from the natural data distribution. Their feature representations exhibit significant variation under image transformations, leading to the failure of adversarial attacks. The proposed IGSA addresses this limitation by enforcing feature-space stability. It ensures that adversarial samples maintain similar feature representations under image transformations, thereby preserving their adversarial efficacy. Next, we provide a formal explanation and experimental validation.

### D.1 FORMAL DEFINITIONS

Let $f : \mathbb{R}^m \to \mathbb{R}^k$ be a classifier decomposed into:

$$f(x) = c(e(x)) \tag{41}$$

where $e : \mathbb{R}^m \to \mathbb{R}^d$ is the feature extractor and $c : \mathbb{R}^d \to \mathbb{R}^k$ is the classifier head.

For natural images $x \sim P_D$ and transformation $T$, we observe:

$$\|e(x) - e(T(x))\|_2 \leq \epsilon_T \tag{42}$$

where $\epsilon_T$ quantifies the model's inherent transformation tolerance.

Traditional adversarial examples $x_{\text{adv}}$ exhibit:

$$\|e(x_{\text{adv}}) - e(T(x_{\text{adv}}))\|_2 \gg \epsilon_T \tag{43}$$

due to their deviation from $P_D$. In contrast, IGSA enhances the stability of adversarial examples in the feature space by increasing their likelihood within the data distribution $P_D$:

$$\|e(x_{\text{adv}}^{\text{IGSA}}) - e(T(x_{\text{adv}}^{\text{IGSA}}))\|_2 \approx \epsilon_T. \tag{44}$$

We provide the following experimental verification.

### D.2 EXPERIMENTAL VALIDATION

#### D.2.1 FEATURE DISTANCE ANALYSIS

Table 17: Feature Space Displacement Under Transformations

| Attack | Blur | Noise | JPEG | Brightness |
|--------|------|-------|------|------------|
| PGD | 18.7 | 22.3 | 15.2 | 12.6 |
| EOT | 14.1 | 17.5 | 11.8 | 9.3 |
| **IGSA (ours)** | **6.8** | **8.4** | **5.1** | **4.7** |

Our experiments demonstrate IGSA's superior stability across transformations. Quantitative analysis using ResNet-50's penultimate layer features shows IGSA achieves 63-73% reduction in feature displacement ($\Delta_{\text{feat}} = \|e(x) - e(T(x))\|_2$) compared to PGD and 52-58% reduction versus EOT, with final displacements ($\Delta_{\text{feat}} \approx 4.7 - 8.4$) approaching natural image variation levels ($\epsilon_T \approx 4.2$). This confirms IGSA's success in maintaining feature-space consistency under perturbations.

