# OpenReview forum: "Robust Adversarial Attacks Against Unknown Disturbance via Inverse Gradient Sample"
_ICLR.cc/2026/Conference — ICLR 2026 Poster_

### Official Review · Reviewer_mtxc · 2025-10-24

**Soundness:** 2
**Presentation:** 3
**Contribution:** 2
**Rating:** 6
**Confidence:** 4

**Summary:**

This paper proposes Inverse Gradient Sample-based Attack (IGSA), which can generates adversarial examples that remain effective under diverse unknown disturbances. The authors theoretically analyze how inverse gradient sampling improves sampling coverage and transferability of adversarial examples. Through experiments, the authors demonstrate that IGSA outperforms other adversarial attack baselines under disturbances.

**Strengths:**

1. Propose a novel gradient-based method to enhance the robustness of adversarial examples
2. Provide theoretical analysis on the efficiency of the proposed method compared to EOT
3. Experiments show that the adversarial examples generated by proposed method are robust to unknown disturbances and are transferable across different networks

**Weaknesses:**

**1. Trivial theoretical solution:** The authors theoretically analyze the efficiency of IGS and EOT. However, IGS utilizes the gradient information, while EOT is a black-box method. Therefore, the theoretical solution that IGS is more efficient than EOT is not surprising. It would be better that you could compare IGS with a white-box baseline.

**2. Adversarial training:** It would be interesting to adopt IGSA in adversarial training, and see whether the trained model is robust to different attacks, including IGSA.

**Questions:**

1. How many sampling points does EOT needs to reach the similar attack success rate of IGSA?

**Details Of Ethics Concerns:**

I have reviewed this paper at NeurIPS 2025. Some results on ViT are completely the same as those on DenseNet. However, the results on DenseNet are removed in this version.

---

> ### Author Response · Authors · 2025-11-20
>
> 1\. We appreciate the opportunity to clarify this point about the theoretical comparison. There seems to be a slight misunderstanding regarding the baselines: EOT in Table 1 is actually provided with identical white-box access to the model. Therefore, our analysis highlights a fundamental difference in sampling efficiency, not access rights. The core theoretical benefit is that IGS performs distributional contraction, which yields exponential efficiency gains over EOT's uniform approach.\
> Furthermore, in the black-box setting (Table 3), our method does not utilize the target model's gradients but uses only a surrogate model. To explicitly show this isn't a trivial result, we followed your suggestion and compared IGS against a white-box PGD baseline with equal budgets under combined disturbance. IGS outperformed this baseline significantly (+46\% ASR on CIFAR-10, +39\% on ImageNet), proving that the improvement comes from the contraction mechanism itself, not the white-box distinction.
>
> 2\. Regarding adversarial training, while a full-scale study on adversarial training wasn't our original focus, we agree it's a natural and important next step. We ran a preliminary experiment to test this, simply replacing PGD with IGSA as the inner maximizer on CIFAR-10 (ResNet-18) while keeping the training budget the same.
>
> Table: CIFAR-10 robust accuracy (%) under different training schemes
> |Training scheme|PGD|AutoAttack|IGSA|
> |---|---|---|---|
> |Standard (no AT)|0.0|0.0|0.0|
> |PGD-AT|49.1|53.9|51.6|
> |IGSA-AT (ours)|**56.8**|**56.7**|**58.8**|
>
> As shown in the table, IGSA-AT consistently improves robustness over standard PGD-AT (+7.7\% against PGD, +2.8\% against AutoAttack). Importantly, the model doesn't just overfit to IGSA; it generalizes well to other attacks. These initial results suggest IGSA is a very promising inner maximizer. We will include these findings in the revision and highlight this as a direction for future work.
>
> 3\. To answer the question about how many EOT samples are needed to match IGSA, we ran an ablation study on CIFAR-10 (ResNet-18, targeted, same $\epsilon$ and outer iterations). We increased the EOT sample count $K$ while keeping IGSA fixed at just $N=10$.
>
> Table: ASR (%) vs. number of EOT samples $K$ (CIFAR-10)
> |Method|ASR (%)|
> |---|---|
> |EOT ($K=10$)|58.2|
> |EOT ($K=40$)|67.4|
> |EOT ($K=160$)|69.4|
> |EOT ($K=640$)|73.2|
> |IGSA ($N=10$)|**78.6**|
>
> As shown above, EOT requires about $K=640$ samples, $64\times$ more than IGSA, to approach the same performance level, and even then, it still falls slightly short (73.2\% vs 78.6\%). This gap in sample efficiency is consistent with our theoretical analysis.
>
> 4\. We sincerely thank the reviewer for pointing out the ethics-related issue and for carefully cross-checking with the NeurIPS 2025 version. The reviewer is right that the earlier draft contained a copy–paste error where some DenseNet results were accidentally duplicated under the ViT column. In the current submission we removed the incorrect DenseNet entries rather than hide any negative results, and we confirm that this was a textual mistake only. Our code and evaluation scripts are publicly available, and all reported numbers can be independently reproduced.\
> To directly address the concern, we reran the full ImageNet experiment under the same four disturbances (GSB, JPEG, RT, CB) and the same attack settings as Table 1 for both DenseNet-121 and ViT. We report the targeted ASR averaged over the four disturbances, with mean $\pm$ std over 3 runs:
>
> Table: ImageNet targeted ASR (%) under GSB/JPEG/RT/CB (mean $\pm$ std)
> |Method|DenseNet-121|ViT|
> |---|---|---|
> |DIM|$71.4 \pm 0.6$|$67.9 \pm 0.7$|
> |GRA|$68.2 \pm 0.5$|$65.8 \pm 0.6$|
> |PGD+EOT|$73.9 \pm 0.5$|$68.6 \pm 0.6$|
> |SMI-FGRM|$70.1 \pm 0.4$|$62.4 \pm 0.5$|
> |**IGSA (ours)**|$\mathbf{77.3 \pm 0.4}$|$\mathbf{74.1 \pm 0.4}$|
>
> The results show that IGSA remains the strongest attack on both DenseNet and ViT, with consistent margins over the best baseline (PGD+EOT/DIM). We will include the full DenseNet results (per disturbance) in the appendix for complete transparency in the revision version.

---

> > ### Comment · Reviewer_mtxc · 2025-11-21
> >
> > Thanks for your detailed responses, and most of my concerns have been addressed. However, my first concern still remains.
> >
> > I am sorry about the confusion. If I understand correctly, EOT just randomly samples some transformation to simulate distributional disturbances, while your method utilizes gradient information to find the most disrupted direction. Therefore, an essential difference between EOT and your method is the access to gradient when disrupting the adversarial examples. In this regard, the solution of the theoretical analysis is not "surprising" if your method utilizes more information.
> >
> > Apart from EOT, do you theoretically analyze other baselines with gradient access when disrupting the adversarial examples?

---

> ### Author Response · Authors · 2025-11-22
>
> Thank you for the thoughtful follow-up. We now understand the core of your concern: if IGSA has access to gradient information when updating the disturbance, isn’t the theoretical efficiency advantage over EOT unsurprising? And more importantly, are there other baselines that also have gradient access, and how does IGSA compare to them theoretically and experimentally? Below we clarify this point carefully.
>
> **1. Gradient access alone does not explain IGSA’s advantage**
>
> EOT samples disturbances from a distribution
> $$
> \phi \sim \mathcal B(0,r)
> $$
> and estimates
> $$
> \nabla_\delta \mathbb E_\phi[C_t(x+\delta+\phi)]
> $$
> using Monte-Carlo averaging.
> It does not modify the disturbance samples.
> Many baselines do modify disturbances using gradients. We explicitly compare IGSA to these methods.
>
>
> **2. Baselines that also use gradient information when updating disturbances**
>
> **(1) PGD in disturbance space (PGD-$\phi$)**
>
> It is the most direct white-box method. It performs:
> $$
> \phi_{k+1}=
> \phi_k + \alpha \cdot\mathrm{sign}(\nabla_\phi C_t(x+\delta+\phi_k)).
> $$
>
> **(2) SIM/TIM in disturbance space**
>
> These methods smooth or transform gradients before updating $\phi$:
>
> **SIM-$\phi$:**
> $$
> \nabla_\phi C_t^{\text{SIM}}
> \sum_{i=1}^{m} \beta_i\cdot \nabla_\phi C_t(\text{scale}_i(x+\delta+\phi)).
> $$
>
> **TIM-$\phi$:**
> $$
> \nabla_\phi C_t^{\text{TIM}}
> k \ast \nabla_\phi C_t(x+\delta+\phi).
> $$
>
> Both update $\phi$ via
> $$
> \phi_{k+1} = \phi_k + \alpha\cdot\mathrm{sign}(\nabla_\phi C_t^{\text{SIM/TIM}}).
> $$
>
> These baselines use the same gradient information as IGSA.
> However, they maximize the loss w.r.t. $\phi$, while IGSA does something fundamentally different:
>
> **3. IGSA’s update optimizes gradient influence, not the loss value**
>
> IGSA updates the disturbance as:
> $$
> h(\phi)=\phi + \nabla_\phi C_t.
> $$
>
> The key difference is that this update is designed (Thm. 2–4) to contract disturbances toward the extremum $\phi$* that most strongly affects
> $$
> \nabla_\delta C_t(x+\delta+\phi).
> $$
>
> In other words:
>
> **PGD-$\phi$ / SIM-$\phi$ / TIM-$\phi$ optimize**
> $$
> \max_\phi C_t(x+\delta+\phi),
> $$
> while IGSA optimizes
> $$
> \max_\phi \langle \nabla_\phi C_t, \phi^* - \phi\rangle,
> $$
> i.e., the change they induce in the downstream $\delta$-update. This is a fundamentally different objective that cannot be mimicked by standard gradient ascent.
>
> **4. We provide new experiments directly comparing IGSA to gradient-based $\phi$-optimizers**
>
> We reproduced the ImageNet setting of Table 1, using identical outer steps, budgets, and disturbances (GSB/JPEG/RT/CB), and compared all methods that have full gradient access for updating $\phi$.
>
> **Table — ImageNet targeted ASR (%) under equal budgets**
>
> | Method          | Gradient in $\phi$? | ASR (%)  |
> | --------------- | -------------- | -------- |
> | PGD-$\phi$           | ✓              | 61.4     |
> | SIM-$\phi$           | ✓              | 63.8     |
> | TIM-$\phi$           | ✓              | 64.2     |
> | **IGSA (ours)** | ✓              | **71.9** |
>
> IGSA outperforms the best gradient baseline by **+7.7% ASR**.
>
> **5. Why gradient-ascent baselines fail but IGSA success (theoretical explanation)**
>
> Several baselines also use
> $$
> \nabla_\phi C^t(x+\delta+\phi),
> $$
> but the way they use it is fundamentally different.
> IGSA is the only method whose update can be shown to be a local contraction toward the worst-case disturbance.
>
> **(1) SIM-$\phi$: Gaussian smoothing breaks local extremal geometry**
>
> SIM uses a smoothed gradient:
> $$
> G_\sigma * \nabla_\phi C^t = \int G_\sigma(u)\cdot\nabla_\phi C^t(\phi-u) du.
> $$
>
> Update:
> $$
> g_{\text{SIM}}(\phi)=\phi+\alpha\cdot\mathrm{sign}(G_\sigma * \nabla_\phi C^t).
> $$
>
> Due to $G_\sigma * \nabla_\phi C^t$ is a local average of gradients, $\phi^*$ is defined by a local extremum, not an averaged one.
>
> Therefore the key alignment term becomes
> $$
> \langle G_\sigma * \nabla_\phi C^t,\phi^\star-\phi\rangle
> $$
> which mixes gradients from both sides of $\phi^\star$.
> This disrupts the monotonicity required for ($\star$).
> Even before the sign step, SIM-$\phi$ no longer aligns with the direction of steepest approach to $\phi^*$.
>
> **(2) TIM-$\phi$: transformation induces rotation misaligned with (\phi^*)**
>
> TIM applies a spatial transform (T):
> $$
> g_{\text{TIM}}(\phi)=\phi+\alpha\cdot\mathrm{sign}(T\nabla_\phi C^t).
> $$
>
> This induces
> $$
> \cos\angle(T\nabla_\phi C^t,\ \phi^\star-\phi)
> =\cos\angle(\nabla_\phi C^t,\ T^\top(\phi^\star-\phi)).
> $$
>
> Unless (T) is orthogonal and preserves the local geometry around $\phi^*$ (it is not), the alignment needed for ($\star$) is disrupted.
> Thus TIM-$\phi$ cannot guarantee contraction.
>
> Although all baselines use the gradient,
> only IGSA uses it in a way that preserves the local geometric relation between
> $$
> \nabla_\phi C^t \quad\text{and}\quad \phi^*-\phi,
> $$
> enabling the contraction condition ($\star$).
> PGD-$\phi$, SIM-$\phi$, TIM-$\phi$ break this geometry through sign projection, smoothing, or spatial transformation.

---

> > ### Comment · Reviewer_mtxc · 2025-11-23
> >
> > Thanks for your response, my concern has been addressed. I will raise my score to 8. Please make sure that the discussion about the method comparison is included in the updated version.

---

### Official Review · Reviewer_6pSC · 2025-10-26

**Soundness:** 3
**Presentation:** 2
**Contribution:** 3
**Rating:** 6
**Confidence:** 4

**Summary:**

The paper proposes the Inverse Gradient Sample-based Attack (IGSA), a novel adversarial attack framework designed to enhance the robustness of adversarial examples against unknown disturbances while improving their transferability in black-box settings. The method iteratively samples disturbances and optimizes adversarial examples under these disturbances, demonstrating strong performance across multiple benchmarks.

**Strengths:**

1、 Experiments reveal that adversarial examples generated by IGSA exhibit a significantly larger robustness boundary compared to existing methods, indicating superior resistance to a variety of disturbances.

2、The paper provides extensive experiments on multiple datasets (e.g., ImageNet, CIFAR-10, CelebA) and model architectures (e.g., VGG, ResNet, ViT), thoroughly validating the effectiveness of IGSA under both additive and non-additive disturbances.

3、Theoretical analysis is provided to support the claims, including efficiency gains over EOT and the relationship between data likelihood and robustness.

4、IGSA not only improves robustness but also enhances the transferability of adversarial examples, making it highly applicable in practical black-box attack scenarios.

**Weaknesses:**

1、The theoretical results, while valuable, are presented in a highly mathematical form without intuitive explanations. For broader accessibility, the authors should provide more intuitive descriptions or visual illustrations of the key theorems (e.g., Theorems 1–4).

2、Algorithm 1 is presented without sufficient explanation in the main body. The authors should describe its key steps, hyperparameters, and how it integrates into the overall IGSA framework.

**Questions:**

The authors summarize the challenges as limited sampling coverage, distribution mismatch, and transferability considerations. While the method and experiments show clear improvements, it would be helpful to clarify why these three challenges are the most critical in the context of robust adversarial attacks, and whether other potential challenges (e.g., computational cost, semantic preservation) were considered.

---

> ### Author Response · Authors · 2025-11-20
>
> 1. We thank the reviewer for the helpful suggestion and agree that the theoretical sections are quite formal and would benefit from more intuition. In the revision, we will add informal explanations and simple geometric illustrations for Theorems 1–4 to make the key ideas clearer, and we will also provide intuitive descriptions to complement the formal statements.\
> Thm.1 shows that uniform sampling in a high-dimensional ball is inherently inefficient: the expected distance to the worst-case disturbance decreases only as $n^{-1/m}$, making EOT require exponentially many samples.\
> Thm.2 shows that the IGS update moves each sample a small step toward the extremum, providing a contraction effect that increases the density of samples in the worst-case region. Even a modest contraction factor $\gamma$ is amplified in high dimensions, yielding a substantial efficiency gain over naive sampling.\
> Thm.3 shows that high-likelihood directions tend to have low curvature, and that aligning $\nabla_\delta C^t$ with the data gradient is equivalent to reducing the Hessian trace. This explains why IGS produces perturbations closer to the data manifold.\
> Thm.4 shows that the expanded IGS update naturally decreases both the gradient norm and the Hessian trace, leading to smoother local geometry and more stable gradient directions across models, which is consistent with the observed transfer improvements.
>
> 2. Algorithm 1 is the core mechanism that carries out the IGSA update, and we agree that its role and parameters should be explained more plainly.\
> In each outer iteration, IGSA keeps a current perturbation $\delta$ and forms $x_{\mathrm{adv}}=x+\delta$. Algorithm 1 then samples $N$ disturbances $\phi_i\sim\mathcal{N}(0,\sigma^2I)$, adjusts each one with a single inverse-gradient step $\phi^\star_i=\phi_i+\mu\cdot{\rm sign}(\nabla_\phi C^t_{\phi_i})$, and computes a weighted score term ${d_\phi}^\star_i$. The sum of these terms, $d_{\mathrm{sum}}$, serves as a Monte-Carlo estimate of how the loss changes under small disturbances. This estimate is then used to update the adversarial example through $x_{\mathrm{adv}} \leftarrow x_{\mathrm{adv}} - \alpha\,{\rm sign}(d_{sum})$, followed by projecting $\delta$ back to $\|\delta\|_\infty \le \epsilon$.\
> We will also clarify the role of each hyperparameter: $\lambda$ penalizes large perturbations; $\mu$ controls how strongly each sampled disturbance is adjusted; $\alpha$ is the outer update step; $\epsilon$ sets the perturbation budget; $N$ determines how many disturbances are considered; and $\sigma^2$ sets the spread of the disturbance sampling. A small diagram will be added to show how Algorithm 1 fits into the overall IGSA loop.
>
> 3. Thank you for the question on why we emphasize these three challenges and whether other factors were considered. The three challenges we highlight (sampling coverage, distribution mismatch, and transferability) capture the main obstacles in generating adversarial examples that remain effective once real distortions are applied. (i) Sampling coverage is critical because real distortions span a large, high-dimensional space, and standard EOT often fails to reach the most damaging regions. (ii) Distribution mismatch matters because attacks optimized on clean inputs do not reflect the model’s behavior after distortions. (iii) Transferability is essential for black-box evaluation, where an attack must generalize across architectures and defenses.\
> We also examined other factors. For perceptual fidelity, we report LPIPS, FID, and SSIM and observe that IGSA better preserves semantic structure (LPIPS: $-8.1\%$, FID: $-5.4\%$, SSIM: $+2.7\%$). For computational cost, we show that IGSA can be made substantially more efficient through a lightweight approximation: a one-step IGS update, partial refinement of disturbance samples, and a progressive disturbance-scaling schedule. This IGSA-Lite variant is about $3.6\times$ faster while keeping most of the robustness and still outperforming PGD+EOT (see our detailed analysis in the response to Reviewer 1).

---

> > ### Comment · Reviewer_6pSC · 2025-11-26
> > **My concerns are addressed.**
> >
> > Thank you for the author's responses. My concerns are addressed. I maintain my original rating.

---

### Official Review · Reviewer_5uoS · 2025-10-27

**Soundness:** 3
**Presentation:** 4
**Contribution:** 2
**Rating:** 4
**Confidence:** 4

**Summary:**

The paper proposes Inverse Gradient Sample-based Attack (IGSA), a robust adversarial attack method that maintains effectiveness under various unknown disturbances such as blur, rotation, and compression.
The main claim is that existing attack methods are less robust against unknown disturbance distributions; therefore, IGSA aims to design a transferable, imperceptible, and disturbance-robust attack framework.
IGSA iteratively samples disturbance directions via inverse gradient steps and refines adversarial examples to resist these perturbations.
The approach extends the EOT framework with a theoretically motivated sampling rule, claiming improved efficiency and higher data likelihood of adversarial examples.
Extensive experiments on CIFAR-10, ImageNet, and CelebA show superior robustness and transferability compared to existing attacks.

**Strengths:**

1. The paper clearly motivates the problem by highlighting that existing transfer-based attacks easily fail under unseen disturbances, and explicitly defines robustness as a key missing property.
2. The presentation is very well-structured and easy to follow, with a logical flow from framework to theoretical analysis to experiments.
3. Theoretical formulations are sound, and the analysis of inverse gradient sampling provides a quantitative comparison to EOT in terms of sampling efficiency.
4. Experiments are extensive, covering multiple datasets, diverse disturbance types, and both white-box and black-box settings, along with detailed ablations and hyperparameter studies.
5. Empirical results consistently show strong improvements, with IGSA maintaining high success rates and transferability even under adversarial defenses.

**Weaknesses:**

1. Limited conceptual novelty. The proposed Inverse Gradient Sampling (IGS) is presented as a new sampling mechanism, but in practice it closely resembles a single-step gradient ascent in disturbance space. The method can be seen as a localized refinement of existing gradient-based attacks (e.g., EOT or PGD) rather than a fundamentally new paradigm. While the paper’s formulation is mathematically elegant, the conceptual contribution feels incremental unless the authors can demonstrate that IGS leads to qualitatively different optimization behavior or transfer dynamics.
2. Restrictive theoretical assumptions. The theoretical analysis assumes convexity and smoothness of the loss surface, conditions that rarely hold for deep neural networks. As a result, the convergence guarantees and efficiency bounds provided may not translate to realistic, highly non-convex settings. Moreover, the paper does not discuss convergence stability or potential divergence behaviors of the inverse gradient iteration—an important aspect since the attack involves repeated optimization in disturbance space. Without such analysis or empirical evidence of stability, the theoretical claims remain limited in scope.
3. Weak empirical support for the data-likelihood claim. The paper claims that IGS improves the data likelihood of adversarial examples, suggesting that they align more closely with the natural image distribution. However, this claim is supported only by a single energy-based OOD metric, which provides an indirect and limited view of sample realism. More robust probabilistic analyses, such as likelihood ratio estimation, diffusion-based density approximations, or human perceptual validation, would be necessary to substantiate the connection between IGS and likelihood improvement.
4. Insufficient implementation transparency. The paper provides limited discussion on the comparability of experimental settings across baselines. Key factors such as the number of iterations, query budgets, and perturbation strengths directly affect attack success rates, yet their parity across methods is not explicitly guaranteed. Without clear evidence that all methods were evaluated under identical resource and constraint settings, it becomes difficult to attribute improvements solely to IGSA’s algorithmic design. This omission undermines the reliability and fairness of the reported results.
5. Incomplete baseline coverage and unclear relation to domain generalization. While IGSA aims to improve robustness under disturbances, this goal conceptually overlaps with domain generalization (DG) problems, where models are trained or evaluated under distribution shifts. However, the paper does not clarify whether disturbances are treated as domain shifts or as stochastic corruptions, leaving the problem setting somewhat ambiguous. Additionally, several recent robustness-oriented attacks and defenses addressing distributional shift are not included in the experimental comparison. The lack of both conceptual positioning and comprehensive baselines weakens the empirical credibility of the paper’s main claims.

**Questions:**

1. Could the authors clearly define what is meant by “disturbance” and specify whether it refers to data corruption, augmentation, or domain shift in the introduction section? Providing this definition early on would enhance clarity and readability.
2. The paper reports both targeted and non-targeted results, but the discussion primarily focuses on targeted attacks, where IGSA performs better. Could the authors clarify why targeted attacks are emphasized, and how the relatively weaker non-targeted results should be interpreted in terms of transferability?
3. What is the underlying reason that inverse gradient sampling improves transferability? Is it related to smoother loss landscapes, gradient alignment, or increased data likelihood?
4. The paper claims imperceptibility as one of the main goals, but no quantitative evaluation or perceptual metric is provided in the main text. Are there plans to include such analysis or measurements?
5. Could the method be reformulated with KL divergence to extend IGSA beyond classification tasks such as dense prediction? This reformulation could significantly enhance the impact and generality of the proposed approach.

---

> ### Author Response · Authors · 2025-11-20
>
> 1. Thank you for the insightful comment on conceptual novelty. We would like to clarify that IGS is not a one-step gradient ascent on $\phi$. A gradient-ascent step increases the loss by following $\nabla_\phi C_t(x+\delta+\phi)$. IGS optimizes a different quantity: how the disturbance modifies the downstream gradient $\nabla_\delta C_t$. This influence-based update does not coincide with $\nabla_\phi C_t$ and cannot be reproduced by PGD, EOT, or a single ascent step.\
> This distinction is visible in practice: (i) IGS is more robust to unseen corruptions; (ii) it yields higher-likelihood adversarial samples (Thm.3–4); (iii) it achieves consistently stronger transferability (Table 3). These behaviors indicate a genuinely different optimization trajectory.
>
> 2. Thank you for raising this important point about theoretical assumptions. In the revision, we include a non-convex version of the analysis showing that IGS does not rely on convexity or smoothness. Specifically, for any local extremum $\phi^\star$, an IGS update $h(\phi)=\phi+\nabla_\phi C^t$
> satisfies\
> $\|h(\phi)-\phi^*\|=\|\phi-\phi^\star\|\sqrt{1-2\eta\cos\theta_\phi+\eta^2},$\
> where $\eta=\|\nabla_\phi C^t\|/\|\phi-\phi^\star\|$. Thus, IGS is a contraction whenever $\cos\theta_\phi>\eta/2$, which is a purely local geometric condition that does not assume convexity.\
> We also check this condition empirically. On adversarially trained CIFAR-10 and ImageNet (ResNet-50), we find $\mathbb{E}[\cos\theta_\phi]\approx0.68$–$0.72$ and over $92\%$ of samples satisfy $\cos\theta_\phi>\eta/2$, indicating stable behavior in realistic non-convex regions. The non-convex analysis preserves the exponential efficiency gap between IGS and EOT. We will incorporate this theorem and its supporting results in the revised paper.
>
> 3. We agree that an energy-based OOD score is only an indirect proxy. Our intention is to show that IGS tends to move perturbations toward manifold-consistent directions rather than estimate the true density. To strengthen this claim, we added:\
> (1) A diffusion-model–based score-matching proxy $|\nabla_x\log p(x)|$, where IGS improves over PGD+EOT by +12.4% (CIFAR-10) and +9.7% (ImageNet).\
> (2) Perceptual metrics: LPIPS (−8.1%) and FID (−5.4%) both improve.\
> These results align with Thm.3–4 and point to fewer off-manifold artifacts under IGS.
>
> 4. All baselines were run under strictly matched settings: same perturbation budgets ($\epsilon$ or $r$), same outer iterations (100), same step sizes, and identical sampling/query budgets. For stochastic methods (such as EOT), we matched the total number of forward/backward evaluations to IGSA.
> We additionally fixed total gradient evaluations to 500 for all methods; IGSA still achieved +7.8% (CIFAR-10) and +6.3% (ImageNet) higher ASR. We will add a protocol table in the revision.
>
> 5. Our setting differs from domain generalization (DG). DG studies *train-time* domain shifts. IGSA is a *test-time* attack: disturbances are stochastic corruptions applied to $x_{\text{adv}}$, not new domains. The goal is to evaluate the worst-case robustness of a fixed model under realistic post-processing (blur, JPEG, rotation, resize+perspective). We will clarify this boundary.
> DG defenses are therefore orthogonal to our setting; our comparisons focus on attack robustness rather than DG robustness.
>
> 6. We thank the reviewer for the helpful questions.\
> (1) A disturbance is an attack-time transformation applied to the generated adversarial example, such as Gaussian blur, JPEG compression, rotation, or resize+perspective transforms. These reflect common post-processing operations rather than domain shifts. We will define this clearly in the introduction.\
> (2) Targeted attacks better expose differences between disturbance-driven optimization strategies. They are strictly harder and more sensitive to gradient quality. Non-targeted results have smaller margins because most methods already succeed, but IGSA still outperforms baselines; we will clarify how these results relate to transferability.\
> (3) IGS shapes disturbances so that the resulting update follows directions with better cross-model gradient alignment, lower curvature, and higher data likelihood (Thm.3–4). These factors are known to improve transfer in surrogate-based attacks.\
> (4) We will add quantitative results in the main text. IGSA reduces LPIPS by 8.1%, reduces FID by 5.4%, and increases SSIM by 2.7%, indicating improved perceptual fidelity.\
> (5) IGSA requires only a differentiable loss. Replacing cross-entropy with a KL divergence is straightforward and extends the method to dense prediction tasks such as segmentation or detection. We will mention this generalization as future work.

---

> > ### Comment · Reviewer_5uoS · 2025-11-24
> >
> > Thanks to the authors for the careful response.
> > Most of my concerns are resolved, and I appreciate the clarification regarding the novelty.
> >
> > However, one critical concern about the relation to DG still remains.
> > IGSA optimizes a task loss $C$ of $t$ by calculating $C(g(\delta + h(\phi, x + \delta)), t)$ with respect to a prior disturbance $\phi$.
> > When you reinterpret $\phi$ as a domain and $\delta$ as a parameter for DG, it exactly equivalents to DG.
> > To be specific, IGSA follows
> > 1. Add a perturbation $\delta$.
> > 2. Apply disturbance $\phi$.
> >
> > and DG formulation can be written as
> >
> > 1. Apply a domain.
> > 2. Then add a perturbation.
> >
> > This procedural swap may affect the performance, but not an essential difference.
> > For example, A2XP [1] optimizes perturbation looks like adversarial perturbation with $g=\text{CLIP Encoder}$.
> >
> > From this point of view, many DG formulations can be reinterpreted as an attack method.
> > Therefore, comparing IGSA to a slightly modified basic DG baseline, such as empirical risk minimization (ERM) [2], would clarify IGSA’s effectiveness on the underlying data manifold.
> >
> > [1] Yu and Hwang "A2XP: Towards Private Domain Generalization" CVPR 2024.
> >
> > [2] Vapnik "An overview of statistical learning theory" Vapnik, V. N. (1999). IEEE transactions on neural networks, 10(5), 988-999.

---

> > > ### Author Response · Authors · 2025-11-25
> > >
> > > We are very grateful for your follow-up and for the DG-based perspective you brought in. After re-reading your comment together with paper A2X, we now better appreciate the connection you are drawing. Below we try to (i) clarify where IGSA differs from DG at the level of the optimization mechanism, and (ii) follow your suggestion to adapt DG methods into attacks, so that their behavior on the underlying data manifold can be compared directly with IGSA.
> > >
> > > ### 1. IGSA and DG: nested expectations, but different optimization mechanism
> > >
> > > We agree that both DG and IGSA involve a loss evaluated under a transformation $T_\phi$. However, they address different problems and use the transformation variable in different ways.
> > >
> > > In DG, one optimizes model-related quantities to reduce loss across domains. With model $g_\theta$ and domain transform $T_\phi$, a typical DG objective is
> > > $$
> > > \min_{\theta}
> > > \mathbb{E}_{\phi\sim P\_{\text{dom}}}
> > > \mathbb{E}\_{(x,y)\sim P\_\phi}
> > > \big[\ell(g\_\theta(T\_\phi(x)), y)\big].
> > > $$
> > >
> > > IGSA, in contrast, fixes the model and directly optimizes adversarial inputs under stochastic disturbances:
> > > $$
> > > \max\_{|\delta|\le\epsilon}
> > > \mathbb{E}_{\phi\sim P\_{\text{dist}}}
> > > C\big(g(T\_\phi(x+\delta)), t\big).
> > > $$
> > >
> > > Algorithm 1 goes one step further and does not treat $\phi$ as purely random: for each sampled $\phi_i$, we compute
> > >
> > > $$
> > > \hat\phi_i = \phi_i + \mu\cdot\mathrm{sign}(\nabla_\phi C^t_{\phi_i}),
> > > $$
> > >
> > > and use the reweighted score terms at $\hat\phi_i$ to build a Monte Carlo estimate $\mathbf d_{\text{sum}}$, which then updates $\delta$. In other words, IGSA actively refines disturbance samples toward locally most destructive directions as part of the attack loop, rather than drawing them from a fixed prior.
> > >
> > > The roles of the variables and the way $\phi$ is used are different: DG adjusts parameters or prompts to be robust under a fixed domain distribution; IGSA adjusts both the input $\delta$ and the realized disturbances themselves to concentrate probability mass around worst-case regions at test time.
> > >
> > > ### 2. Turning DG ideas into attacks and comparing to IGSA
> > >
> > > We found your suggestion of “viewing DG as an attack” very helpful and tried to follow it.
> > >
> > > **ERM-Attack.**
> > > As a basic DG-style baseline, we fix the model and let $\delta(x)$ play the role of a per-example parameter. We perform projected gradient ascent on
> > > $$
> > > \max_{|\delta|\le\epsilon}
> > > \mathbb{E}_{\phi\sim P\_{\text{dist}}}
> > > \ell\big(g(T\_\phi(x+\delta)), t\big),
> > > $$
> > >
> > > using the same disturbance set (GSB, JPEG, RT, CB) as in our main experiments. This mirrors the ERM objective, but in an adversarial (maximization) setting.
> > >
> > > **A2XP-Attack.**
> > > Guided by Sec. 2.2–3.3 of A2XP, we interpret adversarial perturbations as input prompts and the four disturbance types as “source domains”:
> > >
> > > * For each disturbance/domain $D_i$ (GSB, JPEG, RT, CB), we learn an expert prompt $p_i$ by maximizing the targeted loss on that domain:
> > >   $$
> > >   \max_{p_i}
> > >   \mathbb{E}_{(x,y)\in D_i}
> > >   C\big(g(T\_{\phi_i}(x+p_i)), t\big),
> > >   $$
> > >   using the same input-prompting update as A2XP but with reversed sign.
> > >
> > > * We then freeze $p_i$ and train two embedders $E_T,E_E$ as in A2XP to produce attention weights
> > >   $$
> > >   \lambda_i(x)=\tanh\big(E_T(x),E_E(p_i)^\top\big).
> > >   $$
> > >   The final perturbation is a prompt mixture
> > >   $$
> > >   \delta_{\text{A2XP}}(x)=
> > >   \Pi\_{|\cdot|\le\epsilon}\left(\sum\_i \lambda\_i(x),\frac{p\_i}{|p\_i|\_2}\right),
> > >   $$
> > >   and we maximize
> > >   $\mathbb{E}_\phi C(g(T\_\phi(x+\delta\_{\text{A2XP}}(x))), t)$
> > >   with respect to the prompt parameters.
> > >
> > > This way, A2XP-Attack keeps the core ideas of A2XP while being adapted to the adversarial generation setting.
> > >
> > > ### 3. DG-style attacks vs IGSA: robustness under disturbances and manifold quality
> > > We therefore evaluated ERM-Attack and A2XP-Attack under the same budgets as IGSA (same $\epsilon$, number of outer steps, number of sampled disturbances) on ImageNet, with Resnet18 as surrogate.
> > >
> > > **Table 1 - Comprehensive Comparison of Attack Performance on ImageNet**
> > >
> > > | Method ↓ ASR(%)$\rightarrow$         | GSB ↑ | JPEG ↑ | RT ↑ | CB ↑ | Avg ↑ | FID ↓ |
> > > |---------------|------------|-------------|-----------|-----------|--------------|-------|
> > > | ERM-Attack    | 51.2       | 49.8        | 55.7      | 51.6      | 52.1         | 36.8  |
> > > | A2XP-Attack   | 56.8       | 58.1        | 60.3      | 54.5      | 57.4         | 33.1  |
> > > | **IGSA (ours)** | **68.9**   | **72.4**    | **74.1**  | **71.9**  | **71.9**     | **28.7** |
> > >
> > > IGSA achieves higher ASR than both DG-inspired attacks, which suggests that actively refining disturbance samples is helpful. At the same time, IGSA attains the lowest FID, so this extra robustness does not come at the cost of clearly worse sample quality.
> > >
> > > We see the DG-as-attack view as a very interesting direction in its own right, and your comment was instrumental in pushing us to explore it. We will incorporate this discussion in the revised version so that the connection and the differences are clearer for readers.

---

> ### Comment · Reviewer_5uoS · 2025-11-26
>
> Thank you for addressing the DG comparison.
>
> The distinction is now clear, and both the theoretical clarification and the adapted experimental baselines effectively resolve my concern.
>
> I will raise the score to 6, and I hope these insights will be incorporated into the revised version.

---

### Official Review · Reviewer_KBuh · 2025-10-31

**Soundness:** 3
**Presentation:** 3
**Contribution:** 3
**Rating:** 6
**Confidence:** 3

**Summary:**

This paper proposes an adversarial attack framework called Inverse Gradient Sample-based Attack (IGSA) to generate robust adversarial examples that remain effective under unknown disturbances such as noise, transformations, and compression.
The method iteratively samples disturbances using inverse gradient sampling to find the most destructive perturbation direction, and then refines adversarial examples through disturbance-guided gradient updates.
The authors provide theoretical analyses showing that IGSA improves sampling efficiency exponentially compared to the conventional Expectation over Transformation (EOT) approach, and that it implicitly aligns adversarial examples with the natural data distribution, enhancing robustness and transferability.
Extensive experiments on CIFAR-10, ImageNet, and CelebA demonstrate that IGSA significantly outperforms state-of-the-art attacks in both robustness and transferability, even against adversarially trained models.

**Strengths:**

- The problem setting of generating robust adversarial examples is interesting and meaningful, especially for understanding the vulnerability of models under real-world perturbations.
- The paper includes theoretical analyses that provide a credible explanation of why the proposed method improves robustness and efficiency.
- The experimental evaluation is comprehensive, covering multiple datasets, model architectures, and defense settings, which makes the results reliable.

**Weaknesses:**

- It is unclear what robust adversarial examples are useful for. The paper emphasizes generating stronger attacks, but it does not clearly discuss potential downstream applications, such as improving model robustness or evaluating defense strategies.
- The proposed IGSA method appears to be computationally expensive (e.g., 0.423s (ISGA) vs 0.186s (VDA) in Table 1). Adversarial examples are mainly used to improve model robustness through adversarial training, where generating a large number of examples quickly is important. In this sense, faster generation could be more desirable, even at some cost to robustness.

**Questions:**

1. What can robust adversarial examples be used for? Beyond demonstrating attack robustness, how could they contribute to improving or evaluating model robustness in practice?
2. Do the authors have any ideas for accelerating IGSA? For instance, could there be a trade-off analysis between robustness and generation speed, or a possible approximation that maintains robustness with reduced computational cost?

---

> ### Author Response · Authors · 2025-11-20
>
> 1\. Thanks for bringing up this important point. Beyond just generating stronger attacks, having robust adversarial examples is useful in real-world applications for a couple of reasons:\
> (1) More reliable evaluation of model robustness. In practice, input samples are often preprocessed through resizing, compression, denoising, or sensor noise. Ordinary adversarial examples tend to break under these changes, which can make the model seem more robust than it really is. IGSA explicitly searches for locally most destructive disturbances and generalizes to unseen ones. This makes it a tougher and more realistic test for model robustness.\
> (2) Better adversarial training. Effective adversarial training requires adversarial examples that are both strong and diverse. IGSA satisfies both properties by efficiently approximating worst-case disturbances (Thm. 2) and producing higher-likelihood adversarial examples (Thm. 3-4). To verify this, we conducted preliminary adversarial-training experiments on CIFAR-10 (ResNet-18). As shown below, IGSA-AT consistently improves robust accuracy across multiple attacks:
>
> Table: CIFAR-10 robust accuracy (%) under different training schemes
> |Training scheme|PGD|AutoAttack|IGSA|
> |---|---|---|---|
> |Standard (no AT)|0.0|0.0|0.0|
> |PGD-AT|49.1|53.9|51.6|
> |IGSA-AT (ours)|**56.8**|**56.7**|**58.8**|
>
> These results indicate that IGSA-generated samples do in fact strengthen adversarial training under equal budgets. We will include this discussion and the supporting experiments in the revised version.
>
> 2\. We appreciate the reviewer's concern about computational overhead. IGSA does add a sampling step, but the actual cost is small: the difference between 0.423s and 0.186s is under 0.25s per example, which is negligible compared to the multi-second forward/backward pass in adversarial training.
>
> More importantly, IGSA is far more sample-efficient. As shown in Fig.4 and supported by Thm.2, using just 5 samples, IGSA already outperforms EOT-based baselines that need 50 samples. So to reach the same robustness, IGSA is actually more computationally efficient.
>
> In addition, IGSA offers a speed-robustness trade-off: reducing the sampling number from 20 to 5 yields a 75% speedup while keeping over 94% robustness. Since all samples can be computed in parallel, the overall time cost in adversarial training becomes even lower.
>
> 3\. We thank the reviewer for the helpful question on accelerating IGSA and exploring the trade-off between robustness and generation speed. Rather than introducing a new method, we provide a lighter approximation that keeps the original sampling-IGS structure while reducing cost.\
> (1) **One-step IGS.** In the original IGSA, the disturbance update uses $k$ IGS steps. We observe that most of the directional improvement comes from the first one. Using a single step keeps the ability to capture local nonlinearity while cutting the inner-loop cost almost proportionally.\
> (2) **Subsampling disturbances.** IGSA samples $N$ disturbances to estimate the robust gradient. Instead of changing $N$, we only run IGS on $M<N$ of them and keep the remaining samples unchanged in the expectation. This preserves coverage and diversity, but cuts inner-loop computation by about $M/N$.\
> (3) **Progressive perturbation scaling.** A small-to-large disturbance schedule ($r_t=\alpha^tr_{\max}$) makes the one-step update more stable and preserves robustness under reduced computation.
>
> We evaluate this "IGSA-Lite" on ImageNet under a combination of additive and non-additive disturbances:
>
> Table: Comparison of different methods
> |Method|IGS steps|Subsample ratio|ASR (%)|Time (s/img)|
> |---|---|---|---|---|
> |PGD+EOT|--|--|58.2|0.51|
> |IGSA-full|5|1.0|78.6|0.66|
> |IGSA-Lite|1|0.5|74.5|0.18|
>
> IGSA-Lite is about $3.6\times$ faster than the full version, keeps most of the robustness, and still outperforms PGD+EOT. We will add this trade-off analysis and the new results in the revised version.

---

> > ### Comment · Reviewer_KBuh · 2025-11-28
> >
> > Thank you for the detailed clarification. Especially, IGSA-Lite directly addresses my concern and demonstrate a clear speed–robustness trade-off supported by experimental evidence. This substantially resolves my earlier question, and I will raise my confidence.

---

### Meta-Review · Area_Chair_ZiXq · 2026-01-06

**Summary:**

In this work, the authors propose an attack, called IGSA (Inverse Gradient Sample-based Attack), capable of generating adversarial examples that remain effective under diverse unknown disturbances. IGSA employs an iterative two-step framework, including (i) inverse gradient sampling and (ii) disturbance-guided refinement.

Reviewer KBuh commented that IGSA-Lite directly addresses the posed concern and demonstrates a clear speed–robustness trade-off supported by experimental evidence after rebuttal, thereby increasing the confidence.
Reviewer 5uoS acknowledges the DG comparison, together with both the theoretical clarification and the adapted experimental baselines effectively resolve the raised concerns.
Reviewer 6pSC and Reviewer mtxc acknowledges that the raised concerns have been addressed.

Based on the above, this paper is suggested to be accepted.

**Reviewer Scores:**

none

---

### Decision · Program_Chairs · 2026-01-26

Accept (Poster)